# Inflammation leads through PGE/EP$_3$ signaling to HDAC5/MEF2-dependent transcription in cardiac myocytes

András D Tóth[1,2,3,†] (iD), Richard Schell[1,2,4,†], Magdolna Lévay[2,5], Christiane Vettel[2,5], Philipp Theis[1,2], Clemens Haslinger[1,2], Felix Alban[1,2], Stefanie Werhahn[1,2], Lina Frischbier[1,2], Jutta Krebs-Haupenthal[1,2], Dominique Thomas[6], Hermann-Josef Gröne[7], Metin Avkiran[8], Hugo A Katus[4], Thomas Wieland[2,5,†] & Johannes Backs[1,2,†,*] (iD)

## Abstract

The myocyte enhancer factor 2 (MEF2) regulates transcription in cardiac myocytes and adverse remodeling of adult hearts. Activators of G protein-coupled receptors (GPCRs) have been reported to activate MEF2, but a comprehensive analysis of GPCR activators that regulate MEF2 has to our knowledge not been performed. Here, we tested several GPCR agonists regarding their ability to activate a MEF2 reporter in neonatal rat ventricular myocytes. The inflammatory mediator prostaglandin E$_2$ (PGE$_2$) strongly activated MEF2. Using pharmacological and protein-based inhibitors, we demonstrated that PGE$_2$ regulates MEF2 via the EP$_3$ receptor, the βγ subunit of G$_{i/o}$ protein and two concomitantly activated downstream pathways. The first consists of Tiam1, Rac1, and its effector p21-activated kinase 2, the second of protein kinase D. Both pathways converge on and inactivate histone deacetylase 5 (HDAC5) and thereby de-repress MEF2. *In vivo*, endotoxemia in MEF2-reporter mice induced upregulation of PGE$_2$ and MEF2 activation. Our findings provide an unexpected new link between inflammation and cardiac remodeling by de-repression of MEF2 through HDAC5 inactivation, which has potential implications for new strategies to treat inflammatory cardiomyopathies.

**Keywords** histone deacetylase 5; myocyte enhancer factor 2; p21-activated kinase 2; prostaglandin E$_2$; protein kinase D
**Subject Categories** Cardiovascular System; Immunology

## Introduction

Evidence has been provided that sustained stimulation of receptors on the plasma membrane of cardiac myocytes by distinct mediators including endothelin-1 or α-adrenergic agonists is able to trigger the development of pathological structural changes along with alterations of cardiac gene expression (Backs & Olson, 2006a). A pathognomonic change is the reactivation of the dormant embryonic isoforms of genes regulating cardiac growth, myocardial contractility and Ca$^{2+}$-handling (Backs & Olson, 2006a). In particular, the reactivation of fetal cardiac gene programs is suggested to lead to the progression of cardiac remodeling and heart failure. As those mediators play crucial roles in the activation of fetal gene programs, it seems a promising approach to identify and describe the distinct upstream signaling pathways, which may reveal new targets for therapeutic interventions.

The myocyte enhancer factor 2 (MEF2) transcription factors belong to the family of MADS (MCM1, agamous, deficiens, SRF) proteins (Potthoff & Olson, 2007). In mammals, four different MEF2 proteins exist, named MEF2A, B, C, and D. They bind a common DNA sequence and drive the expression of specific target genes (Potthoff & Olson, 2007). The MEF2 proteins have a crucial role in the proper cardiac development. Due to severe cardiac abnormalities, MEF2A- and MEF2C-null mice exhibit pre- and perinatal lethality, respectively (Lin *et al*, 1997; Naya *et al*, 2002). In addition to their critical role in the embryonic development, the MEF2 isoforms and their splicing variants also govern the stress response in the adult heart by reactivation of fetal cardiac gene programs (van Oort *et al*, 2006; Kim *et al*, 2008; Gao *et al*, 2016). Whereas MEF2 transgenic mice develop

---

1 Department of Molecular Cardiology and Epigenetics, Heidelberg University, Heidelberg, Germany
2 DZHK (German Centre for Cardiovascular Research), Heidelberg/Mannheim, Germany
3 Department of Physiology, Faculty of Medicine, Semmelweis University, Budapest, Hungary
4 Department of Cardiology, Heidelberg University, Heidelberg, Germany
5 Experimental Pharmacology, European Center of Angioscience, Medical Faculty Mannheim, Heidelberg University, Mannheim, Germany
6 Institute of Clinical Pharmacology, Goethe University Frankfurt, Frankfurt, Germany
7 Department of Cellular and Molecular Pathology, German Cancer Research Center, Heidelberg, Germany
8 Cardiovascular Division, King's College London British Heart Foundation Centre of Research Excellence, The Rayne Institute, St Thomas' Hospital, London, UK
*Corresponding author. Tel: +49 6221 56 35271; E-mail: Johannes.backs@med.uni-heidelberg.de
†These authors contributed equally to this work

ventricular dilation and contractile dysfunction, mice lacking the MEF2D gene are protected against fetal gene activation, fibrosis, and cardiac hypertrophy (van Oort *et al*, 2006; Kim *et al*, 2008). Therefore, the activity of MEF2 is tightly controlled. Whereas phosphorylation by mitogen-activated protein (MAP) kinases and acetylation by the histone acetyltransferase p300 enhance the transcriptional activity of MEF2 (Potthoff & Olson, 2007; Wales *et al*, 2014; Wei *et al*, 2017), class IIa histone deacetylases (HDACs; HDACs 4, 5, 7, and 9) physically interact with MEF2 and recruit other repressive epigenetic factors (Backs & Olson, 2006a; Lehmann *et al*, 2014). A protective role of class IIa HDACs in heart failure has been suggested. For example, HDAC5 and HDAC9 prevent cardiac hypertrophy (Zhang *et al*, 2002; Chang *et al*, 2004) and HDAC4 is required for the maintenance of cardiac function during physiological exercise (Lehmann *et al*, 2018). Class IIa HDACs can be phosphorylated by several kinases, including the $Ca^{2+}$/Calmodulin-dependent protein kinase II (CaMKII), protein kinase D (PKD), or G protein-coupled receptor kinase 5 (Backs & Olson, 2006a; Kreusser *et al*, 2014; Lehmann *et al*, 2014; Weeks & Avkiran, 2015). In turn, phosphorylated HDACs are guided by 14-3-3 chaperones from the nucleus to the cytoplasm and MEF2 is released from the inhibition and can activate downstream genes (Backs & Olson, 2006a). On the other hand, there are pathways which protect from the harmful hyper-activation of MEF2 (Lehmann *et al*, 2014). For instance, activation of protein kinase A (PKA) has been shown to inhibit MEF2 activity and moreover to counteract CaMKII-mediated activation of MEF2 (Backs *et al*, 2011; Weeks *et al*, 2017; Lehmann *et al*, 2018).

The upstream activation signal of the HDAC-MEF2 axis often originates from G protein-coupled receptors (GPCRs). The divergent pathways resulting from GPCRs could have antagonistic effects on MEF2 activation; therefore, it is difficult to predict the net effect on MEF2 activation by different mediators. The aim of this study was to identify new mediators that signal via GPCRs in cardiac myocytes and which were not known to activate the HDAC-MEF2 axis before. Furthermore, we sought to explore the detailed resulting downstream signaling pathway. We found that prostaglandin $E_2$ (PGE$_2$), which is one of the main inflammatory mediators, strongly activates MEF2. PGE$_2$ binds to the $G_{i/o}$-coupled EP$_3$ receptor, which activates divergent, parallel operating signaling pathways; one involving Tiam1-, Rac1-, and p21-activated kinases (PAK) and another involving PKD and HDAC5. We found that full MEF2 activation depends on participation of both pathways and that MEF2 is activated *in vivo* in inflamed hearts.

# Results

## PGE$_2$ activates MEF2 through the EP$_3$ receptor

To identify unknown GPCR-dependent signaling pathways that regulate MEF2 activity, we conducted a screening experiment using neonatal rat ventricular myocytes (NRVMs). NRVMs were infected with an adenovirus harboring a MEF2-reporter (3xMEF2-Luc), which responds to endogenous MEF2. Thereafter, the cells were stimulated with different GPCR agonists for 24 h in serum-free medium. Similar to previous reports, endothelin-1 (100 nM) activated MEF2 to a high extent (Backs *et al*, 2011). On the other hand, the β-adrenergic receptor agonist isoproterenol (1 μM) significantly decreased the basal

level of MEF2 activity. We identified several mediators, which slightly elevated the activity of MEF2, such as sphingosine-1-phosphate (1 μM) or lysophosphatidic acid (10 μM). Strikingly, we observed a 20-fold activation by two related compounds, namely prostaglandin E$_1$ (PGE$_1$; 10 μM) and prostaglandin E$_2$ (PGE$_2$; 10 μM; Fig 1A). In contrast, analogs of other prostaglandins (fluprostenol and treprostinil, agonists of prostaglandin F and prostacyclin receptors, respectively) had no significant effect. In the following experiments, we focused on PGE$_2$ because of its higher abundance in the heart *in vivo* (Herman *et al*, 1987). The PGE$_2$ effect on MEF2 activity was concentration-dependent (Fig EV1A). In good agreement with the ability to activate MEF2, PGE$_2$ triggered the expression of known specific MEF2 target genes (Potthoff & Olson, 2007; Lehmann *et al*, 2018), such as *Nur77*, *Myomaxin*, or *Adamts1* (Fig 1B). MEF2 activation is commonly associated with the induction of hypertrophic gene programs. Likewise, PGE$_2$ increased the mRNA levels of the hypertrophy marker *BNP* (Fig 1B) and induced cellular hypertrophy of NRVMs, which correlated with the induction of MEF2 activity (Fig 1C). In addition, we observed a similar extent of protein synthesis induction after PGE$_2$ stimulation as with the well-known hypertrophic α-adrenoceptor agonist phenylephrine (Fig EV1B). Next, we aimed to determine the receptor involved in PGE$_2$-mediated MEF2 activation. The four different isoforms of PGE$_2$ receptors (EP$_1$, EP$_2$, EP$_3$, EP$_4$) show a different degree of G protein coupling and expression patterns (Woodward *et al*, 2011). Each of the four types is expressed in cardiac myocytes, but EP$_3$ and EP$_4$ receptors are the most abundant isoforms (Di Benedetto *et al*, 2008). The EP$_3$ receptor antagonist L798106 inhibited PGE$_2$-dependent MEF2 activation (Figs 1D and EV1C), while neither the EP$_4$ receptor antagonist L-161,982 (2 μM) nor the EP$_1$/EP$_2$ receptor antagonist AH6809 (10 μM) led to an alteration of MEF2 activity, suggesting the EP$_3$ receptor as the underlying receptor to activate MEF2 by PGE$_2$ in NRVMs.

## PGE$_2$ activates MEF2 through $G_{i/o}$ proteins

The EP$_3$ receptor is generally considered to couple to pertussis toxin (PTX)-sensitive $G_{i/o}$ proteins, thereby decreasing intracellular cAMP levels. In accordance, EP$_3$ receptor inhibition by L798106 increased the isoproterenol-induced phosphorylation of phospholamban, a known cAMP-regulated protein (Cuello *et al*, 2007; Fig EV1D). However, the EP$_3$ receptor has several splice variants, which differ in the G protein-coupling properties (Woodward *et al*, 2011). Therefore, we examined which heterotrimeric G protein types are essential for the PGE$_2$-mediated MEF2 activation by a pharmacological and molecular approach (Fig 2A). Regulator of G protein signaling (RGS) proteins are GTPase activating proteins (GAPs) and specifically block different classes of G proteins. RGS16 is a GAP for $G_{i/o}$ and $G_{q/11}$ proteins, RGS2 suppresses $G_{q/11}$, while RGS-LSCII inhibits the $G_{12/13}$ signaling pathway (Vettel *et al*, 2012). Like RGS2, p63ΔN, a specific scavenger of activated $G\alpha_{q/11}$ proteins did not affect the PGE$_2$-induced activity of MEF2. In contrast, the $G_{i/o}$ and $G_{q/11}$ inhibitor RGS16 (Fig 2B), as well as pretreatment with pertussis toxin (PTX, 100 ng/ml) attenuated the PGE$_2$ response (Fig 2C). We did not observe any significant effect of RGS-LSCII overexpression in this setting (Fig 2B). These data suggest that neither $G_{q/11}$- nor $G_{12/13}$-, but the PTX-sensitive $G_{i/o}$ protein activation is required for the PGE$_2$ response. To rule out an involvement of $G_s$ proteins, we inhibited adenylyl cyclase, the effector enzyme of $G_s$ proteins. The adenylyl

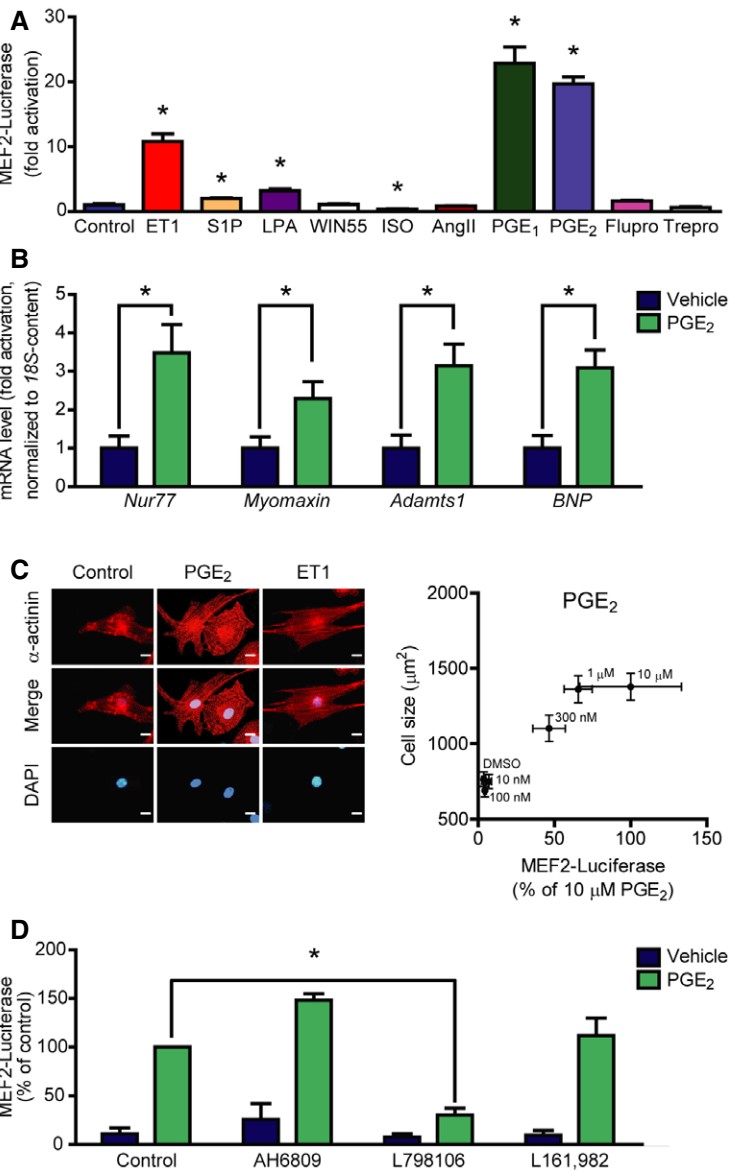

**Figure 1. Screening the effect of different GPCR agonists on MEF2 activity.**

A   Neonatal rat ventricular myocytes (NRVMs) were infected with the 3xMEF2-Luc reporter, serum starved for 20 h, and stimulated for 24 h with different agonists: 100 nM endothelin-1 (ET1), 1 μM sphingosine-1-phosphate (S1P), 10 μM lysophosphatidic acid (LPA), 1 μM WIN55,212-2 (WIN55, cannabinoid receptor agonist), 1 μM isoproterenol (ISO, β-adrenergic receptor agonist), 100 nM angiotensin II (AngII), 10 μM prostaglandin $E_1$ (PGE$_1$), 10 μM prostaglandin $E_2$ (PGE$_2$) or 100 nM prostanoid F receptor agonist fluprostenol (Flupro), 100 nM treprostinil (Trepro, prostacyclin receptor agonist).

B   PGE$_2$ induces the expression of MEF2 target genes. NRVMs were serum starved for 24 h, then were stimulated with 1 μM PGE$_2$ for 2 h, and mRNA levels of the indicated genes were determined.

C   Correlation between MEF2 activation and NRVM hypertrophy. MEF2 activity was measured as in (A) upon increasing concentration of PGE$_2$. Cell size of NRVMs was determined after 48-h stimulation with the indicated concentrations of PGE$_2$. Left: Representative images of α-actinin stained NRVMs stimulated with DMSO, 300 nM PGE$_2$, or 1 nM ET1. Scale bar is 10 μm.

D   PGE$_2$ activates MEF2 via EP$_3$ receptor. NRVMs were infected with the 3xMEF2-Luc reporter and serum starved for 20 h. The cells were stimulated with DMSO or 1 μM PGE$_2$ for 24 h in the presence or absence of different EP receptor antagonists: AH6809 (10 μM, EP$_1$- and EP$_2$-antagonist) or 798106 (200 nM, EP$_3$-antagonist) or L161,982 (2 μM, EP$_4$-antagonist).

Data information: Values are mean ± s.e.m. In (A), the experiment was performed in triplicates, and similar results were obtained in three different experiments. Student's two-tailed *t*-test, *$P < 0.05$ vs. control (ET1, $P = 0.0013$; S1P, $P = 0.0144$; LPA, $P = 0.026$; WIN55, $P = 0.6205$; ISO = 0.0399; AngII, $P = 0.5812$; PGE$_1$, $P = 0.001$; PGE$_2$, $P < 0.0001$; Flupro, $P = 0.0846$; Trepro, $P = 0.1958$). In (B), $n = 5$, technical replicates, Student's two-tailed *t*-test, *$P < 0.05$ vs. control (*Nur77*, $P = 0.0148$; *Myomaxin*, $P = 0.0417$; *Adamts1*, $P = 0.0112$; *BNP*, $P = 0.0065$). In (C), NRVM cell size was determined from a minimum of 100 cells of three technical replicates upon different concentrations of PGE$_2$, and MEF2 activity was parallel assessed from six technical replicates. Correlation was statistically analyzed by calculating Pearson correlation coefficient (Pearson $r = 0.9626$, $P = 0.0021$). In (D), $n = 3$, independent experiments, *represents significant interaction between the two treatments ($P < 0.05$, two-way ANOVA, AH6809, $P = 0.1092$; L798106, $P = 0.0002$; L161,982, $P = 0.5579$).

Source data are available online for this figure.

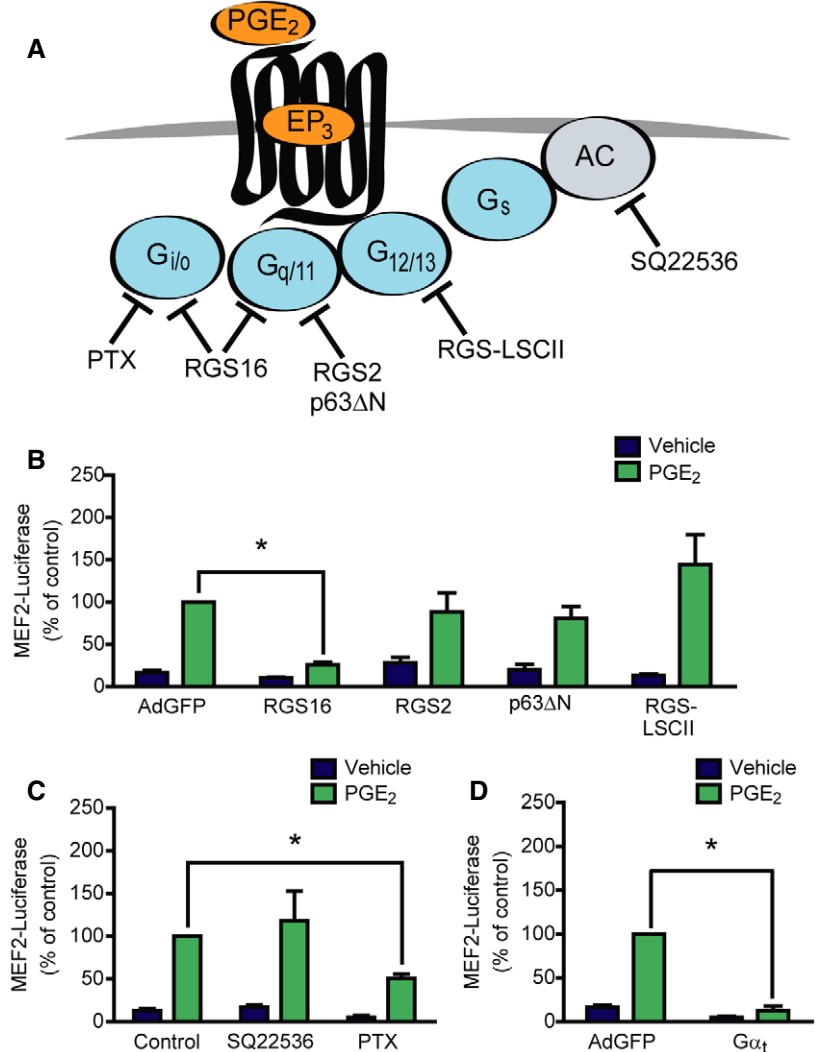

**Figure 2. PGE$_2$-mediated MEF2 activation is G$_{i/o}$-βγ dependent.**

A　　Illustration of the target points of the used pharmacological and protein-based inhibitors.

B–D　(B, D) NRVMs were infected with the 3xMEF2-Luciferase reporter and with recombinant adenoviruses encoding EGFP (AdGFP), RGS16, RGS2, p63ΔN, RGS-LSCII, or Gα$_t$, as indicated. The cells were serum starved for 20 h and stimulated with DMSO or 1 μM PGE$_2$ for 24 h. (C) NRVMs were infected with the 3xMEF2-Luciferase reporter and pretreated with the adenylyl cyclase inhibitor SQ22536 (100 μM) for 20 min or the G$_{i/o}$ protein inhibitor pertussis toxin (PTX, 100 ng/ml) for 20 h in serum-starved conditions and treated with DMSO or 1 μM PGE$_2$ for 24 h.

Data information: In (B–D), $n = 3$, independent experiments, *represents significant interaction between the two treatments ($P < 0.05$, two-way ANOVA, RGS16, $P < 0.0001$; RGS2, $P = 0.1725$; p63ΔN, $P = 0.0666$; RGS-LSCII, $P = 0.0818$, SQ22536, $P = 0.385$; PTX, $P < 0.0001$; Gα$_t$, $P < 0.0001$), values are mean ± s.e.m.
Source data are available online for this figure.

cyclase inhibitor SQ22536 (100 μM) did not affect MEF2 activity (Fig 2C). In addition, since the adenylyl cyclase inhibition also mimics the effect of Gα$_{i/o}$, the absence of alteration suggested that not the α, but the βγ subunit of G$_{i/o}$ is the transducer in the PGE$_2$-mediated MEF2 activation. In accordance, overexpression of Gα$_t$ to scavenge the free Gβγ subunits abolished the PGE$_2$ effect (Fig 2D). Similar results were obtained with the overexpression of long isoform of RGS3 (RGS3L), another Gβγ scavenger (Vogt *et al*, 2007; Fig EV2). Taken together, these data indicate that the PGE$_2$-induced MEF2 activation is mediated by the βγ subunit of G$_{i/o}$ proteins.

**PGE$_2$ induces HDAC5 hyperphosphorylation and de-repression of MEF2 via PKD**

Among critical regulators of MEF2 are class IIa HDACs. A well-known mechanism of de-repression of MEF2 activity is HDAC phosphorylation followed by its concomitant nucleo-cytoplasmic shuttling. Therefore, we analyzed the phosphorylation status of HDAC5 by Western blot. We overexpressed FLAG-tagged HDAC5 in NRVMs. After stimulation with PGE$_2$, we observed a high extent of hyperphosphorylation of HDAC5 at Ser-498 (Fig 3A), similarly to endothelin-1 (ET1, 100 nM) or phenylephrine (PE, 100 μM). Since

Ser-498 is a specific phosphorylation site of protein kinase D (PKD; Haworth *et al*, 2011), we examined whether PKD is responsible for the phosphorylation of HDAC5. Indeed, pretreatment with the specific PKD-inhibitor BPKDi (3 μM; Meredith *et al*, 2010; Monovich *et al*, 2010) diminished the PGE$_2$-induced HDAC5 phosphorylation. In accordance, phosphorylation of PKD was increased at Ser-744/Ser-748 and Ser-916 sites upon PGE$_2$ treatment in NRVMs (Fig 3B), which was partly (phosphorylation at Ser-744/Ser-748) observed in adult mouse ventricular myocytes as well (Fig EV3). Different mechanisms of PKD activation exist. It is known that PKD can be activated in a protein kinase C (PKC)-dependent way, e.g., by PE. In contrast, ET1 stimulates PKD activity in a PKC-independent manner in NRVMs (Vega *et al*, 2004; Harrison

*et al*, 2006; Haworth *et al*, 2007; Guo *et al*, 2011). Thus, the involvement of PKC in the PGE$_2$-mediated PKD and HDAC5 phosphorylation was examined. Phosphorylation of PKD at Ser-744/Ser-748, a well-known PKC-target site, was diminished upon pretreatment with the PKC-inhibitor bisindolylmaleimide I (BIM, 2 μM). However, phosphorylation of the autophosphorylation site (Ser-916) was only slightly decreased (Fig 3B), and HDAC5 phosphorylation at Ser-498 was not altered (Fig 3A) by BIM, suggesting that the activation of PKD by PGE$_2$ occurs both PKC dependently and PKC independently. Nevertheless, BPKDi prevented MEF2 activation by PGE$_2$ (Fig 3C). In contrast, inhibition of another HDAC kinase, CaMKII, by auto-camtide-2-related inhibitory peptide II (AIP, 1 μM) did not affect PGE$_2$-mediated MEF2 activity. We therefore conclude that PGE$_2$

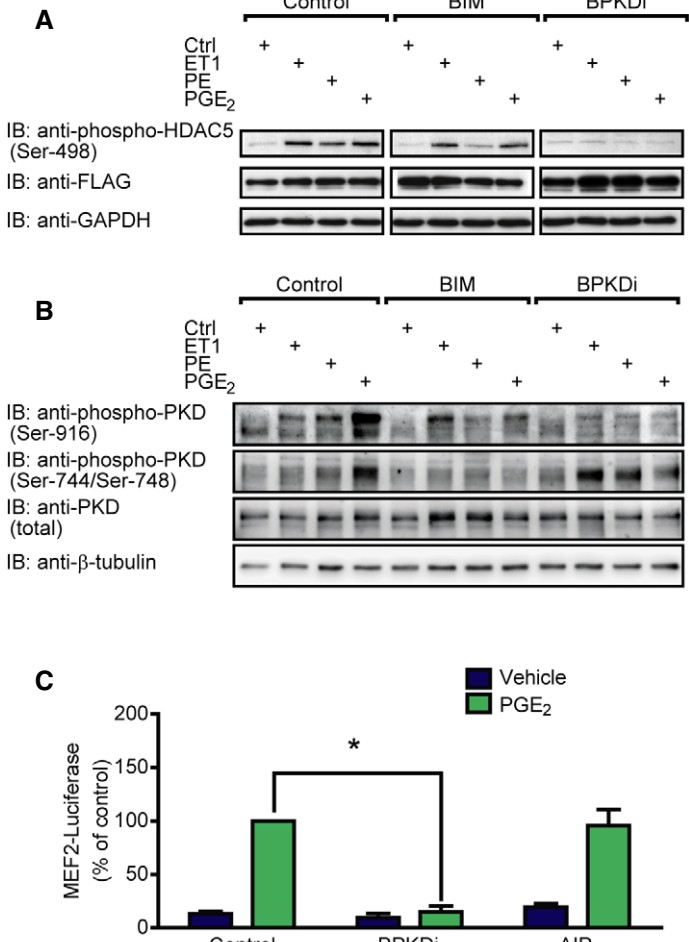

**Figure 3.  PGE$_2$ induces HDAC5 hyperphosphorylation and concomitant MEF2 activation via PKD.**

A, B   NRVMs were infected with recombinant adenovirus encoding Flag-HDAC5 (A) or not infected (B) and serum starved for 20 h. The cells were pretreated with the PKC-inhibitor bisindolylmaleimide I (2 μM) or the PKD-inhibitor BPKDi (3 μM) for 20 min and stimulated with 100 nM endothelin-1 (ET1) or 100 μM phenylephrine (PE) or 1 μM PGE$_2$ for 4 h, as indicated. HDAC phosphorylation was detected by immunoblotting with the anti-HDAC5 phospho-Ser-498 antibody, phosphorylation of PKD was assessed using anti-phospho-PKD antibodies (Ser-744/Ser-748 and Ser-916), and representative images are shown from three independent experiments.

C   NRVMs were infected with the 3xMEF2-Luc reporter and serum starved for 20 h. The cells were pretreated with the PKD-inhibitor BPKDi (3 μM) or the CamKII-inhibitor AIP (1 μM) for 20 min and stimulated with DMSO or 1 μM PGE$_2$ for 24 h. $N = 3$, *represents significant interaction between the two treatments ($P < 0.05$, two-way ANOVA, BPKDi, $P < 0.0001$; AIP, $P = 0.2197$). Values are mean ± s.e.m. The exact $n$ and $P$-values can also be found in the Source Data Excel file for Fig 3.

Source data are available online for this figure.

promotes HDAC5 hyperphosphorylation and de-repression of MEF2 activity via PKD.

### PGE₂ triggers Rac1 activation via Tiam1

Several effector partners of $G_{i/o}$-βγ have been described, among others the small G protein Rac1 and RhoA (Vogt et al, 2007). To test whether these small G proteins are activated by PGE₂ in NRVMs, we performed effector pull-down assays and we found, that exclusively Rac1, but not RhoA is activated after PGE₂ stimulation (Fig 4A). To examine whether Rac1 activation is $G_{i/o}$-dependent, we used PTX treatment and overexpression of RGS16. Both manipulations suppressed the PGE₂-induced Rac1 activation (Fig 4B and C). Earlier studies have shown that the $G_{i/o}$-mediated Rac1 activation in NRVM can occur in a $G_{i/o}$-βγ-, phosphatidyl-inositol-3-kinase (PI3K)-, and the guanine nucleotide exchange factor Tiam1-dependent manner

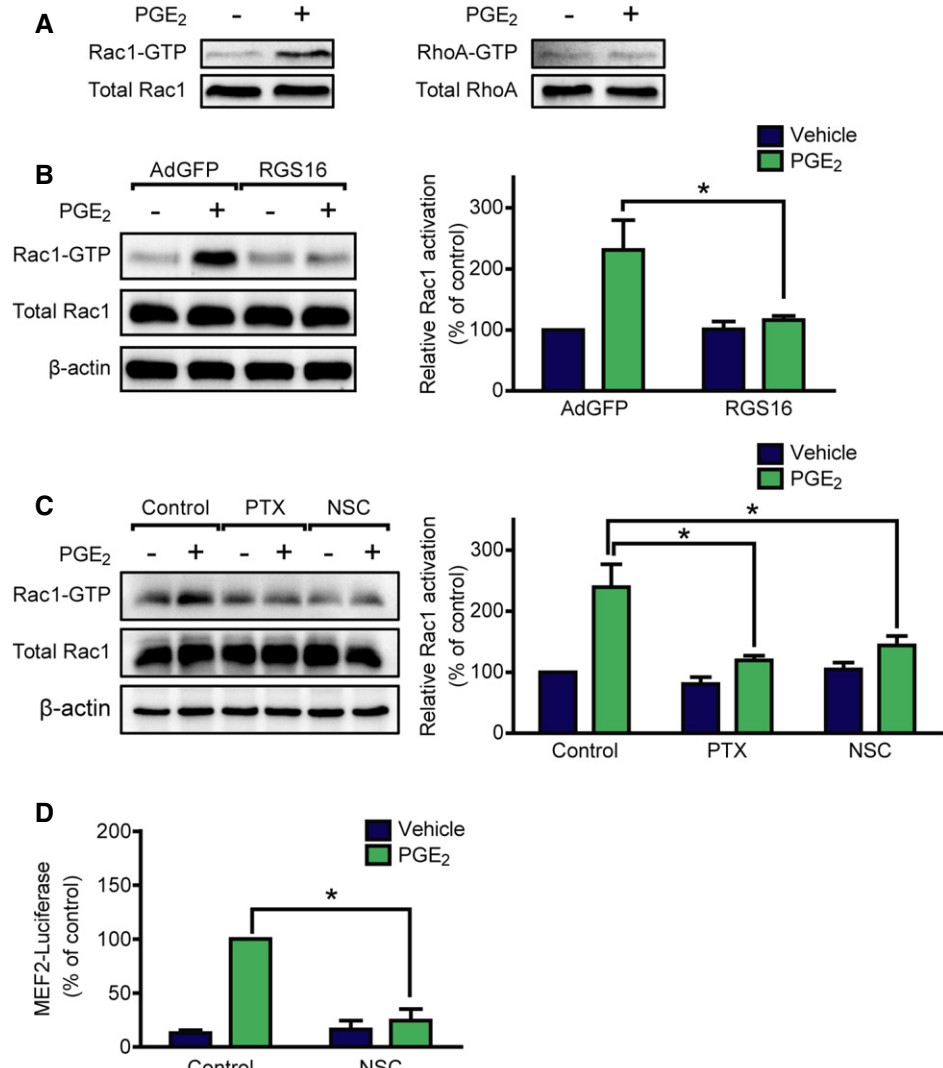

**Figure 4.  PGE₂ induces Rac1 and MEF2 activation via $G_{i/o}$ and the RacGEF Tiam1.**

A   NRVMs were serum starved for 24 h and stimulated for 2 min with 10 μM PGE₂. Rac1 and RhoA GTPase activation was determined by effector pull-down assay and analyzed by immunoblot.

B   NRVMs were infected with recombinant adenoviruses encoding EGFP (AdGFP) or RGS16 24 h prior to stimulation, and then, Rac1 activity was measured.

C   Cells were pretreated with 100 ng/ml pertussis toxin (PTX, $G_{i/o}$-inhibitor) for 24 h or with 50 μM NSC23766 (NSC, Tiam1–Rac1 interaction inhibitor) for 2 h. Thereafter, Rac1 activation was determined by effector pull-down assay.

D   NRVMs were transduced with the 3xMEF2-Luc reporter and were serum starved for 20 h. After 2 h of 50 μM NSC23766 (A) pretreatments, the cells were stimulated with DMSO or 1 μM PGE₂ for 24 h, and then, MEF2 activity was determined. NSC abolished the MEF2 activation by PGE₂.

Data information: In (A–C), representative images from at least three independent experiments are shown. In (B–D), values are mean ± s.e.m., (B, C) $n = 4$; (D), $n = 3$; *$P < 0.05$ (two-way ANOVA, RGS16, $P = 0.0415$ (B); PTX, $P = 0.039$ (C), NSC effect on Rac1 activation, $P = 0.0412$ (C); NSC effect on MEF2 induction, $P < 0.0001$. PGE₂ effect was significant in Rac1 activation, $P < 0.0001$). The exact $n$ and $P$-values can also be found in the Source Data Excel file for Fig 4.
Source data are available online for this figure.

and that the interaction between Tiam1 and Rac1 can be inhibited by NSC23766 (Vettel *et al*, 2012). Treatment with 50 μM NSC23766 (NSC) diminished the PGE$_2$-induced Rac1 activation (Fig 4C), suggesting that PGE$_2$ triggers Rac1 activation via G$_{i/o}$ and Tiam1.

### PGE$_2$-induced MEF2 but not PKD activation requires Tiam1-dependent Rac1 activation

We hypothesized that MEF2 activation by PGE$_2$ may thus occur downstream of Tiam1 and Rac1. For verification, we treated NRVMs with NSC (Fig 4D). The Tiam1–Rac1 interaction inhibitor diminished the MEF2 activation after PGE$_2$ stimulation, indicating that the PGE$_2$-induced signaling pathway toward MEF2 includes Tiam1 and Rac1. To investigate the theoretical possibility that Rac1 exerts its effect on MEF2 through PKD, we measured PKD phosphorylation after treatment with NSC (Fig 5A). However, PGE$_2$-induced PKD phosphorylation at Ser-744/Ser-748 and Ser-916 was not affected by NSC treatment. Likewise, NSC could not prevent PGE$_2$-induced HDAC5 phosphorylation at Ser-498 (Fig 5B). Moreover, inhibition of PKD did not block the Rac1 activation (Fig EV4). p21-activated kinases (PAKs) are well-characterized effector proteins of Rac1 (Vettel *et al*, 2012). In accordance, the phosphorylation of the isoform 2 of PAK (PAK2) was increased upon PGE$_2$ at Ser-192/Ser-197 and Thr-402 sites in NRVMs in a partly EP$_3$ receptor-dependent manner but not in cardiac fibroblasts, similar to the phosphorylation of PKD (Fig 5C). Thus, we examined their role in MEF2 regulation, as possible participants in the Rac1-induced MEF2 activator pathway. Indeed, the PAK inhibitor IPA-3 (30 μM) significantly attenuated the activation of MEF2 upon stimulation with PGE$_2$ (Fig 5D). We then tested whether the Tiam1–Rac1–PAK2 pathway regulates the cellular localization of HDAC5. Like ET1, PGE$_2$ induced the nucleo-cytoplasmic shuttling of HDAC5 (Fig 6). Strikingly, IPA-3 decreased the cytosolic translocation of HDAC5, similar to the PKD-inhibitor BPKDi, which was even stronger through a combined pretreatment with both compounds. These findings suggest that PGE$_2$ promotes more likely two parallel pathways regulating nucleo-cytoplasmic shuttling of HDAC5. The first is dependent on PKD activation and HDAC phosphorylation, and the other involves the Tiam1–Rac1-mediated activation of PAK2 affecting nucleo-cytoplasmic shuttling of HDAC5 by a mechanism that remains to be identified. Thus, two pathways converge on HDAC5 and may promote MEF2 activation synergistically.

### Cardiac inflammation leads to increased myocardial PGE$_2$ levels and MEF2 activation *in vivo*

To confirm the *in vivo* relevance of these results, we used the LPS-induced endotoxemia model to induce myocardial inflammation. The inflammatory effect in this model was confirmed by increased mRNA levels of pro-inflammatory cytokines including interleukin 6 (*IL6*) and tumor necrosis factor α (*TNFα*; Fig 7A). Importantly, the LPS-treated mice displayed more than twofold higher levels of PGE$_2$ in the myocardium (Fig 7B). Histological and "whole organ" staining of β-galactosidase activity in MEF2-lacZ reporter mice showed a lacZ-positive myocardium and in particular cardiac myocytes as sign for MEF2 activation in cardiac myocytes, which was not observed in sham-treated mice (Fig 7C). Moreover, we found increased activity of the *in vitro* identified pathways upstream to MEF2 in

inflammation. Phosphorylation of PKD at Ser-744/Ser-748 and of HDAC5 was significantly elevated, and we observed a slight but statistically significant increase of PAK2 phosphorylation at Thr-402 (Fig 7D).

## Discussion

Our results identify PGE$_2$ as a powerful activator of the MEF2 transcription factor in cardiac myocytes. PGE$_2$ is one of the main mediators of inflammation (Woodward *et al*, 2011). Several studies linked chronic inflammation with the progression of cardiac remodeling and heart failure but the exact mechanism how signals are transduced toward cardiac gene expression are not understood (Hofmann & Frantz, 2013). Since the role of MEF2 is well established in transcriptional activation that drives adverse cardiac remodeling, our findings reveal a yet unrecognized link between myocardial inflammation and adverse remodeling leading to heart failure and unmask potential new drug targets in the setting of inflammatory cardiomyopathies. Accordingly, several groups have shown that PGE$_2$ induces cardiomyocyte hypertrophy but could not link their results to a distinct transcriptional pathway (Mendez & LaPointe, 2005; Frias *et al*, 2007; He *et al*, 2010). In these studies, pro-hypertrophic effects of the G$_{i/o}$-coupled EP$_3$ and the G$_s$-coupled EP$_4$ receptors were reported. Here, we show that the EP$_3$, but not EP$_4$ receptor is responsible for the induction of MEF2 activity, suggesting that EP$_4$ signals to another transcriptional pathway that still needs to be identified.

Phosphorylation of class IIa HDACs is a critical signaling event for their nucleo-cytoplasmic shuttling and de-repression of MEF2 activity (Vega *et al*, 2004). Here, we show that PGE$_2$ leads to activation of PKD and HDAC5 phosphorylation at its 14-3-3 binding site that induces nucleo-cytoplasmic shuttling. HDAC5 phosphorylation was not influenced by CaMKII inhibition, which is consistent with our previous results that CaMKII selectively phosphorylates HDAC4 but not HDAC5 (Backs *et al*, 2006b, 2008). On the other hand, data of our group suggested a role of CaMKII for leukocyte recruitment via a newly identified cardiac myocyte-intrinsic immune response (Weinreuter *et al*, 2014), suggesting the existence of another CaMKII-HDAC4-dependent pathway in cardiac inflammation that is independent of the here described PGE$_2$-EP$_3$-PKD-HDAC5 axis.

Protein kinase D is considered as a powerful mediator of myocardial contraction, cardiac hypertrophy, and remodeling (Cuello *et al*, 2007; Avkiran *et al*, 2008; Fielitz *et al*, 2008). Interestingly, the properties of PKD activation originating from different GPCRs can show distinct characteristics in cardiac myocytes. For example, the α$_1$-adrenergic receptor agonist phenylephrine only activates PKD in a PKC-dependent manner and the activated form rapidly shuttles from the sarcolemma to the nucleus. In contrast, ET1 can activate PKD via PKC-dependent and PKC-independent signaling pathways and promotes more sustained activation of PKD at the sarcolemma (Bossuyt *et al*, 2011). We found that PGE$_2$, like ET1, can activate PKD also through a PKC-independent mechanism. Thus, we propose that PGE$_2$ stimulates PKD via G$_{i/o}$-βγ. Gβγ was demonstrated to activate PKD directly by interaction with its pleckstrin homology domain (Jamora *et al*, 1999). In addition, Gβγ was also shown to hinder HDAC5 activity through direct binding, which might add to the effects described in this study (Spiegelberg & Hamm, 2005).

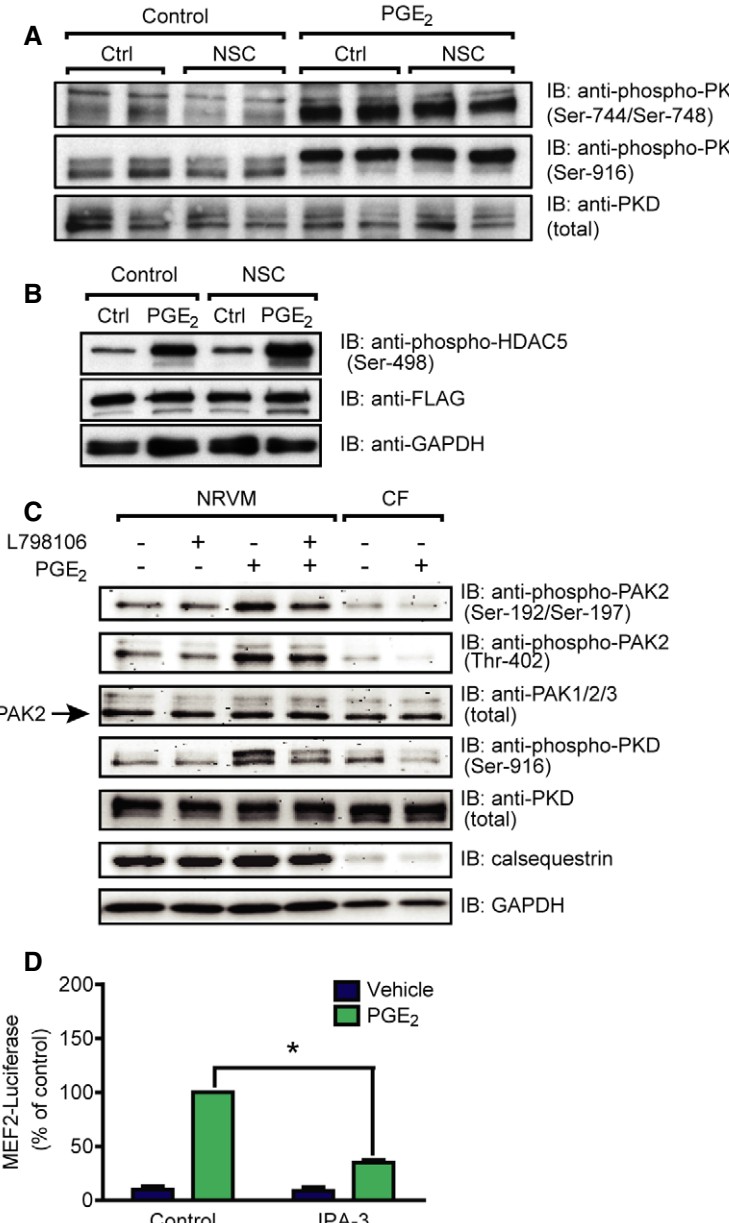

**Figure 5. PGE2 activates MEF2 through PAK2 in NRVMs.**

A  After serum starvation, NRVMs were pretreated with or without 50 μM NSC23766 for 2 h and stimulated with 1 μM PGE2 for 1 h. Total and phosphorylated PKD levels were analyzed by immunoblot. PKD was activated by PGE2, but it was not affected by NSC.

B  After infection with Flag-HDAC5 and serum starvation, NRVMs were treated with 50 μM NSC23766 for 2 h prior to stimulation. The cells were stimulated with 1 μM PGE2 for 1 h. HDAC5 phosphorylation at Ser-498 was detected by immunoblot. The phosphorylation of HDAC5 at this site was not prevented by NSC.

C  NRVMs and neonatal cardiac fibroblasts (CFs) were serum starved, pretreated with vehicle or 200 nM L798106 (EP3 receptor antagonist) for 20 min, and were stimulated with 1 μM PGE2 for 10 min. PAK2 and PKD phosphorylation were assessed by immunoblot. Calsequestrin was used as a myocyte marker.

D  MEF2 activity was determined in cells infected with the 3xMEF2-Luc reporter after serum starvation. Cells were stimulated with vehicle or 1 μM PGE2 for 24 h after 1-h of 30 μM IPA-3 pretreatment. IPA-3 inhibited the MEF2 activation by PGE2.

Data information: In (A–C), representative images from three independent experiments are shown. In (D), values are mean ± s.e.m., $n = 3$, biological replicates, *$P < 0.05$ (two-way ANOVA, $P < 0.0001$). The exact $n$ and $P$-values can also be found in the Source Data Excel file for Fig 5.
Source data are available online for this figure.

Monomeric GTPases of the Rho family have been associated with regulation of HDAC function. For example, activation of RhoA leads to HDAC5 phosphorylation (Harrison *et al*, 2006). G$_{i/o}$-coupled receptors are able to activate several RhoGTPases, including RhoA and Rac1 in cardiac myocytes (Vogt *et al*, 2007). However, we only could detect increased Rac1 activity after PGE2 stimulation. Rac1 is apparently indispensable for the hypertrophic response of cardiac myocytes (Pracyk *et al*, 1998; Vettel *et al*, 2012). Moreover, Rac1 activation also

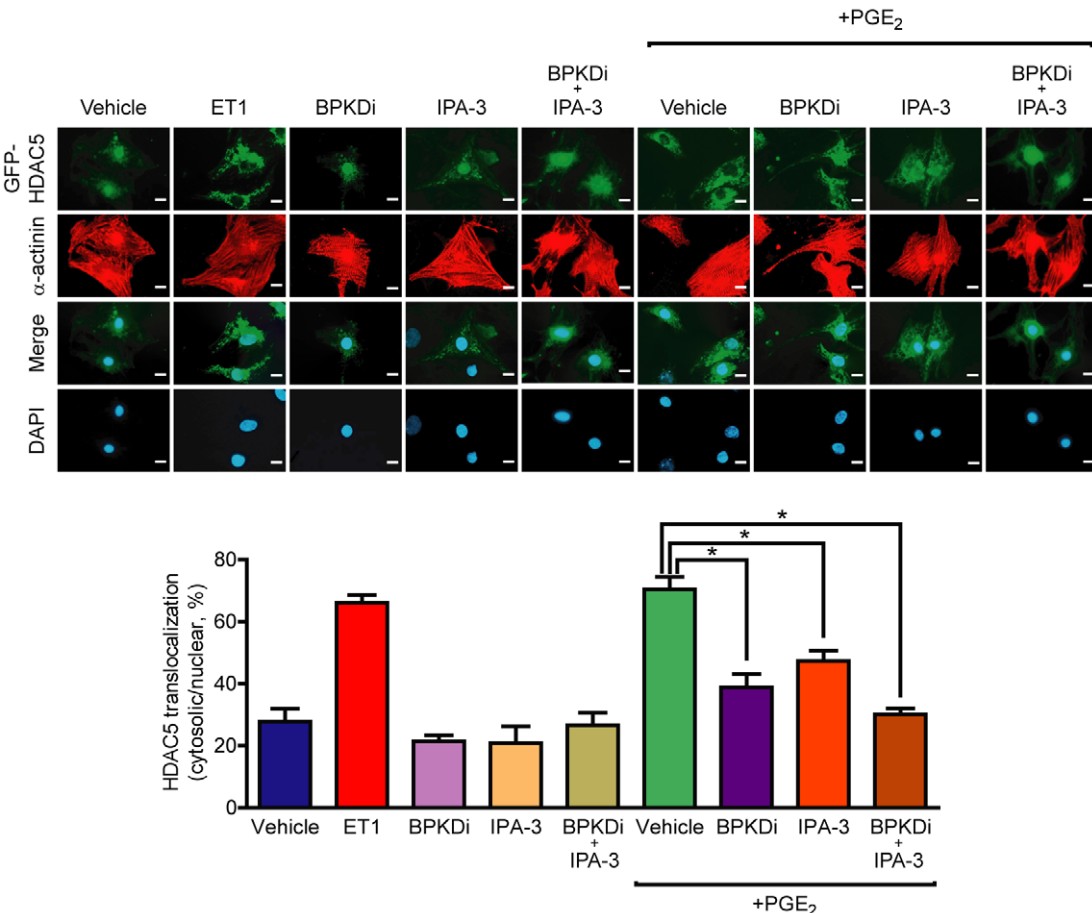

**Figure 6. PGE2-induced nuclear export of HDAC5 is regulated by both PKD and PAK2.**

Top: GFP-HDAC5-infected NRVMs were serum starved for a day, pretreated for 2 h with the indicated compounds (BPKDi, 3 μM, IPA-3, 30 μM) alone or in combination, then were stimulated for 4 h. Representative images. Scale bar is 10 μm. Bottom: Statistical analysis of cellular localization of HDAC5. GFP-HDAC5 localization (nuclear or cytosolic) was assessed in > 100 cells of each sample ($n = 3$, technical replicates). *$P < 0.05$ (one-way ANOVA with Bonferroni *post hoc* test, vehicle vs. ET1 or vs. $PGE_2$, $P < 0.0001$; $PGE_2$ vs. $PGE_2$ + BPKDi, $P = 0.0004$; $PGE_2$ vs. $PGE_2$ + BPKDi, $P = 0.0114$; $PGE_2$ vs. $PGE_2$ + BPKDi + IPA-3, $P < 0.0001$). The exact $n$ and $P$-values can also be found in the Source Data Excel file for Fig 6.

Source data are available online for this figure.

promotes oxidative stress by activation of NADPH oxidase and contributes to cardiac injury and malfunction (Maack *et al*, 2003; Ma *et al*, 2013). Rac1 signaling was also shown to lower class I HDAC activity in cardiac myocytes (Ma *et al*, 2013). Earlier, we reported that stimulation of $\alpha_{1A}$-adrenergic receptors promotes Rac1 activation, which is required for myocyte hypertrophy. This Rac1 activation was mediated by $G_{i/o}$-βγ/PI3K signaling and by the widely expressed RacGEF Tiam1 (Vettel *et al*, 2012). Similarly, we found that the $PGE_2$-mediated Rac1 activation depends on Tiam1 and the Tiam1–Rac1–PAK pathway was essential for MEF2 activation. We found that, similar to PKD, PAK2 activation contributes to nucleo-cytoplasmic shuttling of HDAC5 upon $PGE_2$ stimulation. These findings suggest that simultaneous activation of at least two different pathways converge on HDAC5 and subsequently MEF2. Whether and where PAK2 phosphorylates HDAC5 directly or indirectly remains however to be determined. Thus, we propose a model, in which $PGE_2$-dependent MEF2 activation requires nucleo-cytoplasmic shuttling of HDAC5 by PKD and Rac1-PAK2 in parallel (Fig 8). The necessity of multiple signaling pathways for the

complete activation of MEF2 is in good agreement with the results of previous studies. For instance, nuclear export of HDACs can be blocked without altering their phosphorylation, and acetylation of MEF2 was shown to be essential for its complete activation (Wei *et al*, 2017). In earlier studies, $PGE_2$ has also been shown to elevate the intracellular cAMP level in cardiac myocytes through the $EP_4$ receptor (Liu *et al*, 2012). We and others have demonstrated before that increased cAMP levels after β-adrenergic receptor stimulation counteracts MEF2 activation (Backs *et al*, 2011; Haworth *et al*, 2012; Chang *et al*, 2013; Lehmann *et al*, 2014, 2018). Interestingly, $PGE_2$ increases cAMP concentration in different cellular compartments than β-adrenergic receptors (Liu *et al*, 2012). We speculate that the β-adrenergic receptor-induced cAMP pool may be more efficient to repress MEF2 than the $PGE_2$-induced pool. Since PKA activation was associated with both beneficial and deleterious outcomes in cardiac myocytes (Lehmann *et al*, 2014), it would be valuable to determine the exact cAMP compartment responsible for MEF2 repression because its pharmacological manipulation may offer advantageous impact.

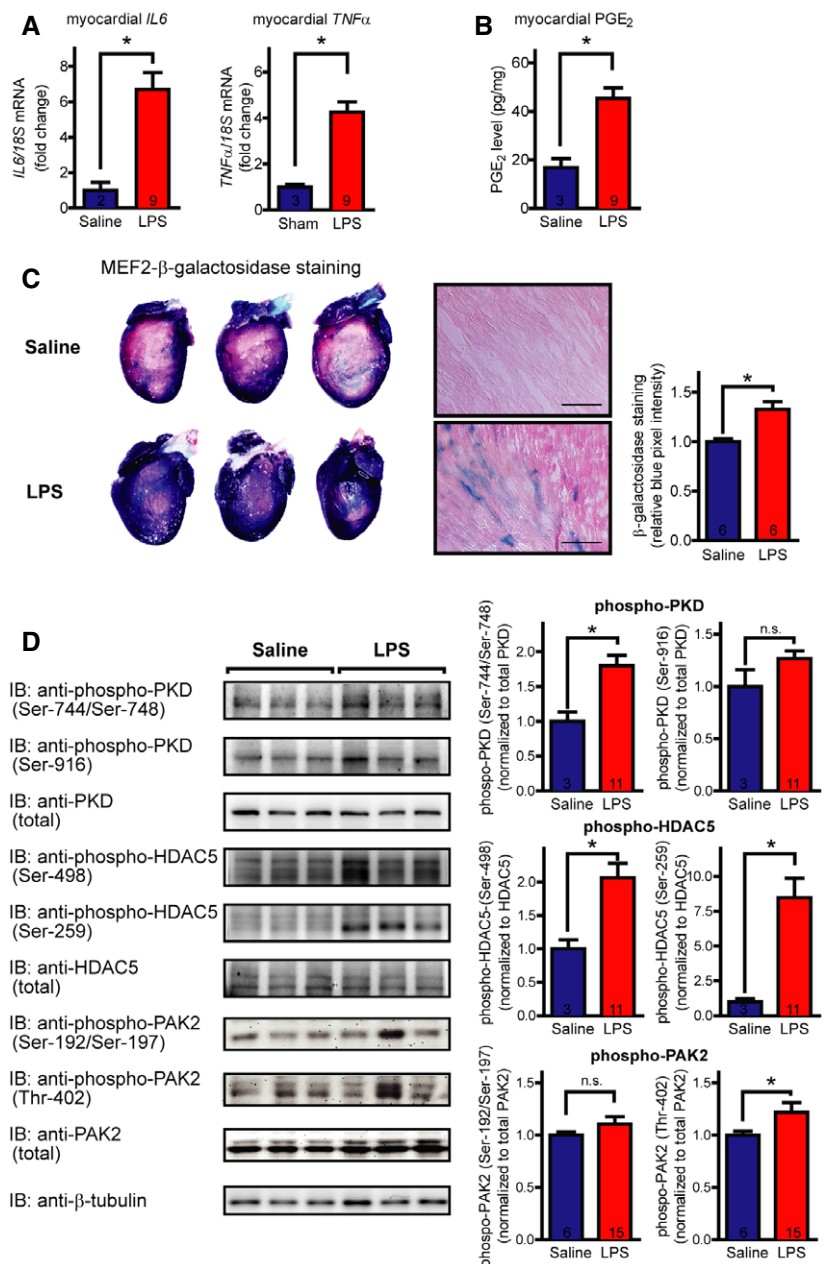

**Figure 7.  LPS-induced endotoxemia leads to myocardial inflammation, increased myocardial PGE$_2$ levels, and MEF2 activation.**

MEF2-lacZ reporter mice (BALB/c background, 6–12 weeks old) were treated with 7 mg/kg lipopolysaccharide from *Escherichia coli* (O111:B4) or saline intraperitoneally and were sacrificed after 24 h.

A  mRNA levels of different inflammatory cytokines, as indicated. The graphs show relative mRNA levels, fold increase compared to saline-treated controls, normalized to 18s content.

B  PGE$_2$ was quantified after mechanical homogenization of deeply frozen hearts using nano-liquid chromatography-tandem mass spectrometry.

C  Histologic and macroscopic stainings of β-galactosidase activity in MEF2-lacZ reporter mice show MEF2 activation (blue cells and precipitates, respectively) in the myocardium upon LPS treatment. Saline-treated littermates served as control. Scale bar of histological stainings is 100 μm. Quantification of whole-heart stainings is shown in the right panel (pixel intensity in blue channel normalized to total intensity).

D  Induction of PKD, HDAC5, and PAK2 in myocardial inflammations. Immunoblots were performed on extracts of hearts of saline- and LPS-treated mice. Representative blots are shown in the left panel, and the quantification is in the right panel.

Data information: Values are mean ± s.e.m. The exact *n* values are shown in the bottom of the bar graphs. *$P < 0.05$, Welch's two-tailed unpaired *t*-test was used for statistical analysis, n.s. = not significant. In (C), representative images of myocardial sections are shown from three independent experiments and representative whole hearts of six independent experiments. *P*-values: IL6, $P = 0.0007$ (A); TNFα, $P < 0.0001$ (A); PGE$_2$, $P = 0.0016$ (B); blue pixel intensity, $P = 0.0083$ (C); phospho-PKD-Ser-744/Ser-748, $P = 0.0049$ (D); phospho-PKD-Ser-916, $P = 0.2684$ (D); phospho-HDAC5-Ser-498 (D), $P = 0.0019$; phospho-HDAC5-Ser-259, $P = 0.0003$ (D); phospho-PAK2-Ser-192/Ser-197, $P = 0.1962$ (D); phospho-PAK2-Thr-402, $P = 0.0446$ (D). The exact *n* and *P*-values can also be found in the Source Data Excel file for Fig 7.
Source data are available online for this figure.

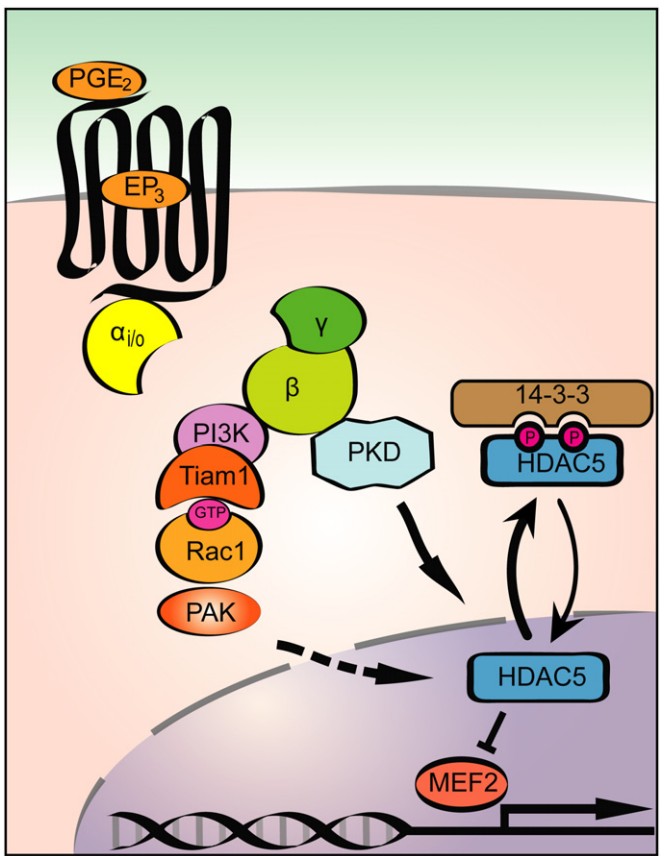

**Figure 8.  Divergent pathways promoted by PGE₂ synergistically activate the transcription factor MEF2.**

Schematic representation of the PGE₂ triggered signal transduction pathways converging on HDAC5 and subsequently MEF2. PGE₂ activates MEF2 through EP₃ receptor and the βγ subunit of G$_{i/o}$ proteins. One pathway implicates activation of PKD and concomitant phosphorylation of HDAC5. The phosphorylated HDAC5 is exported from the nucleus, which results in de-repression of MEF2 and activation of the embryonic gene programs. In addition, hand in hand activation of Tiam1, Rac1, and PAK2 kinase occurs, which pathway also contributes to the nuclear export of HDAC5 and is responsible for synergistic activation of MEF2.

**Therapeutic implications**

Besides the *in vitro* data, we provide *in vivo* evidence that cardiac inflammation in a sepsis model induces activation of MEF2. Thus, the unmasked upstream signaling pathway consisting of PGE₂, EP₃, Rac1, and PKD may provide a number of new drug target candidates for the treatment of heart failure in the specific setting of inflammatory causes. Cyclooxygenase inhibitors, such as acetylsalicylic acid, are generally used as anti-inflammatory, anti-nociceptive, and anti-thrombotic agents (Woodward *et al*, 2011). However, these drugs inhibit the synthesis of all prostaglandin types, and some of them, besides others prostacyclin, were shown to have beneficial effects in cardiovascular homeostasis (Woodward *et al*, 2011). A more specific approach could be the inhibition of microsomal prostaglandin E synthases (mPGES), the responsible enzymes for PGE₂ production. However, mice lacking mPGES show worse cardiac function after angiotensin II infusion and develop adverse

cardiac remodeling in an experimental model of myocardial infarction (Degousee *et al*, 2008; Harding *et al*, 2011). Cardiac-specific EP₄$^{-/-}$ mice suffer from reduced cardiac function after MI and dilated cardiomyopathy develops with aging in EP₄$^{-/-}$ male mice (Qian *et al*, 2008; Harding *et al*, 2010). These results indicate that PGE₂ may play an important role in maintaining the cardiac homeostasis partly through EP₄ signaling. On the other hand, overexpression of EP₃ receptors induces cardiac hypertrophy in the murine heart (Meyer-Kirchrath *et al*, 2009). A role of EP₃ receptor was also demonstrated in the promotion of vascular neointimal hyperplasia and formation of thrombocyte aggregation (Woodward *et al*, 2011; Zhang *et al*, 2013). In patients with severe heart failure, PGE₂ levels are significantly higher, indicating the hyper-activation of the prostaglandin system (Dzau *et al*, 1984). Thus, our findings imply that EP₃ receptors represent a potential target in the treatment of inflammatory cardiomyopathy. Moreover, the identified downstream molecules of EP₃ such as PKD or Rac1/PAK2 could also represent novel drug targets for inflammatory cardiomyopathies, for which no specific therapeutic strategies are available yet (Heymans *et al*, 2009; Suthahar *et al*, 2017). Further *in vivo* investigations using, e.g., EP₃-, PKD-, or MEF2D-deficient mice in the context of inflammatory cardiomyopathies are warranted to determine a critical role of the above mentioned signaling pathways and may facilitate translational studies toward personalized strategies to combat inflammatory cardiomyopathies.

# Materials and Methods

### Chemical reagents

Phenylephrine (PE), endothelin-1 (ET1), dibutyryl-cAMP, AH6809, L-798106, isoproterenol, and bicinchoninic acid protein assay kit were purchased from Sigma-Aldrich. Prostaglandin E1, prostaglandin E2, SQ 22536, and treprostinil were purchased from Santa Cruz, autocamtide-2-related inhibitory peptide II, L-161,982, bisindolylmaleimide, and pertussis toxin from Calbiochem. Sphingosine-1-phosphate, lysophosphatidic acid, IPA-3, and NSC23766 were from Tocris. Fluprostenol was from Cayman, and WIN55,212-2 was from Enzo. Angiotensin II was bought from American Peptide, and the Luciferase Assay Kit was purchased from Promega. BPKDi was custom-synthesized (Haworth *et al*, 2012). [4,5-³H]-leucine was from Perkin-Elmer. Cell culture reagents were from Invitrogen.

### Cell culture and adenoviral infection

Neonatal rat ventricular myocytes (NRVMs) were isolated from 1- to 2-day-old Sprague Dawley rats as previously described (Backs *et al*, 2006b). After isolation, NRVMs were maintained in DME/199 medium (4:1) with 10% FBS, 2 mM L-glutamine, 100 U/ml penicillin, and 100 μg/ml streptomycin. NRVMs were infected 48 h after plating, grown 24 h in serum-free media. Adenoviruses (Ad) harboring FLAG-HDAC4, FLAG-HDAC5, 3xMEF2-Luc, GFP, RGS2, RGS16, p63ΔN, RGS-LSCII, RGS3L, and transducin α were described before and used with 10 MOI (Fahimi-Vahid *et al*, 2002; Vogt *et al*, 2007; Backs *et al*, 2008, 2011; Zhou *et al*, 2008; Wuertz *et al*, 2010; Carbajo-Lozoya *et al*, 2012; Vidal *et al*, 2012). The adenovirus encoding GFP-HDAC5 was a kind gift from Eric N. Olson. Neonatal

cardiac fibroblasts (CFs) were separated from cardiomyocytes by pre-plating. A cell suspension equal to two neonatal rat hearts was seeded per well of a 6-well plate. After 1 h, the supernatant was removed, and the surface was rinsed with culture media to remove cell debris and suspended cells. The attached cells were cultured for 2–3 days to confluence and subsequently deprived of serum 24 h prior to stimulation. Adult mouse ventricular myocytes were isolated as previously described (Kreusser et al, 2014). Briefly, hearts of 6- to 12-week-old wild-type BALB/c mice were excised after isoflurane anesthesia, which were then retrogradely perfused and digested. The cells were plated and incubated on laminin-coated plates for 16 h, then were serum starved for 4 h prior to stimulation.

### Pharmacological characteristics of the used inhibitors

Detailed description of the compounds and their use can be found in Appendix Table S1.

### MEF2 reporter assay

NRVMs were infected with Ad–3xMEF2-Luc to measure endogenous MEF2 activity. 4 h after infection, the cells were serum starved for 20 h. After starvation, cells were stimulated for 24 h. Thereafter, the cells were harvested, and luciferase activity was detected by Promega Luciferase Assay Kit and BMG Labtech FLUOstar OPTIMA luminometer. For standardization, the samples' protein content was determined by Bicinchoninic Acid Protein Assay Kit according to manufacturer's instruction.

### Immunoblotting

After infection and/or stimulation, NRVMs and adult mouse ventricular myocytes were washed and harvested in a solution containing 150 mM NaCl, 50 mM Tris, pH 7.4, 1 mM EDTA, 1% Triton X-100, 1 mM PMSF, and protease inhibitors (Complete; Roche). To determine PAK and phospholamban phosphorylation, NRVMs and neonatal cardiac fibroblasts were harvested in Kranias buffer (30 mM Tris–HCl, pH 8.8, 5 mM EDTA, 30 mM NaF, 3% SDS, 10% glycerol supplemented with protease inhibitor cocktail and phosphatase inhibitor cocktail). Extracts from cardiac tissue were obtained as previously described (Kreusser et al, 2014). The lysates were denatured, solved in Laemmli buffer, and electrophoresed with SDS–PAGE, then transferred to a PVDF membrane using wet transfer. The membrane was blocked in a 5% skim milk/PBS solution containing 0.1% Tween-20 (PBST) or Roti-Block (Carl Roth) solution (for detection of phosphorylated proteins) for 30 min. The membrane was then probed with the primary antibody overnight at 4°C. After washing three times with PBST, the membrane was incubated with the secondary antibody for 1 h at room temperature, followed by washing with PBST again. The signal was detected using ECL Advance Western Blotting System (GE Healthcare). The antibodies used for immunoblotting were as follows: rabbit anti-phospho-HDAC5 (Ser-498; ab47283, 1:1,000, Abcam), rabbit anti-phospho-HDAC5 (Ser-259; produced and provided by Timothy A. McKinsey, 1:1,000), rabbit anti-HDAC5 (#20458, 1:1,000, Cell Signaling), mouse anti-Flag M2-peroxidase (HRP, 1:5,000, Sigma), mouse anti-GAPDH (MAB374,

1:5,000, Chemicon, Millipore), mouse anti-RhoA, (26 C4, 1:200, Santa Cruz), mouse anti-Rac1 (610650, 1:1,000, BD Transduction), rabbit anti-PKD/PKCµ [#90039, 1:1,000, Cell Signaling (Huynh & McKinsey, 2006)], rabbit anti-phospho-PKD/PKCµ (Ser-744/748; #2054, 1:1,000, Cell Signaling), rabbit anti-phospho-PKD/PKCµ (Ser-916; #2051, 1:1,000, Cell Signaling), rabbit anti-calsequestrin (#PA1-913, 1:2,000, ThermoScientific), rabbit anti-PAK1/2/3 (#2604, 1:1,000, Cell Signaling, used in NRVM and CF experiments), rabbit anti-PAK2 (#2615, 1:1,000, Cell Signaling, used for mouse heart lysates), rabbit anti-phosho-PAK1/2 (Ser-199/Ser-204/Ser-192/Ser-197; #2605, 1:1,000, Cell Signaling), rabbit anti-phospho-PAK1/2 (Thr-423/Thr-402; #2601, 1:1,000, Cell Signaling), rabbit anti-phospholamban (Ser-16; #A010-12, 1:5,000, Badrilla), mouse anti-phospholamban (#A010-14, 1:5,000, Badrilla). HRP-conjugated secondary antibodies (goat anti-rabbit and goat anti-mouse, 1:5,000) were from Bio-Rad.

### RhoGTPase activation assay

NRVMs were adenovirally infected or pretreated with the indicated inhibitors before the RhoGTPase activation assay. The cells were lysed in GST-Fish buffer. GTP-bound RhoA was precipitated with the Rho-binding domain of Rhotekin, and the GTP-bound Rac1 was precipitated with the Rac-binding domain of PAK1 coupled to glutathione sepharose, as previously described (Vettel et al, 2012). To determine the quantity of GTP-bound and total GTPases, immunoblot analysis was performed.

### [³H]-Leucine incorporation assay

NRVMs were serum starved and treated 1 µCi/ml [4,5-³H]-leucine containing media. The [³H]-incorporation was determined as previously described (Vettel et al, 2012). Briefly, after thorough washing with ice-cold PBS, the proteins were precipitated with 10% trichloroacetic acid for 4 h at 4°C. Following repeated washing with ice-cold PBS, the pellet was dissolved with 0.1 M NaOH and 0.01% SDS for 2 h at 37°C. Thereafter, the samples were mixed scintillation liquid from Carl Roth GmbH. [³H]-scintillation was measured with a Tri-Carb 2100TR Packard scintillation counter.

### Immunostaining and analysis of HDAC5 cellular localization

NRVMs were grown on collagen-coated coverslips. For cell size measurements, the cells were serum starved for 24 h and were stimulated with the indicated stimuli for 48 h; the stimulation was repeated after first 24 h. To determine the cellular localization of HDAC5, NRVMs were transduced with GFP-HDAC5, and the medium was changed the next day to starvation medium for 24 h. The cells were pretreated for 2 h and stimulated for 4 h with the indicated compounds. After stimulation, the cells were washed, fixed in 4% paraformaldehyde, permeabilized using 0.3% Triton X-100, and blocked in 5% goat serum-containing PBS solution. The cells were stained with mouse anti-α-actinin (sarcomeric) primary antibody (clone EA-53, 1:400, Sigma) and Alexa Fluor 594-conjugated goat anti-mouse IgG secondary antibody (#A-11005, 1:1,000, Molecular Probes). DAPI was added in 1:10,000 dilution. Images were taken with an Olympus BX-51 fluorescence microscope using 20× or 63× objectives. Cell size was analyzed with ImageJ software.

Intracellular (i.e., nuclear or cytosolic) localization of GFP-HDAC5 was determined in more than 100 cells in each sample. The investigator was blinded.

### Induction of endotoxemia

Six- to twelve-week-old male and female BALB/c mice, harboring a MEF2-lacZ reporter, were treated with 7 mg/kg bacterial lipopolysaccharide (LPS) from *Escherichia coli* (O111:B4, Sigma-Aldrich), as described by others (Rudyk *et al*, 2013). Littermates received saline and served as a control, and the mice were randomly allocated. Sacrificing and organ harvesting was performed after 24 h. The MEF2-lacZ reporter consists of a lacZ transgene linked to the hsp68 basal promoter and three tandem copies of the MEF2 site from the desmin enhancer and the MEF2-lacZ reporter mice were a kind gift of Eric N. Olson (Naya *et al*, 1999; Passier *et al*, 2000). Animals were kept in a 12-h/12-h light/dark cycle at 21–23°C and fed with a standard chow diet *ad libitum* before experiments. All experimental procedures were reviewed and approved by the Institutional Animal Care and Use Committee at the Regierungspräsidium Karlsruhe, Germany.

### Analysis of myocardial PGE$_2$ levels

Directly after harvest, murine hearts were frozen in liquid nitrogen and mechanically homogenized without intermittent thawing. Concentrations of PGE$_2$ were determined using liquid chromatography-tandem mass spectrometry (LC-MS/MS) as described previously (Sisignano *et al*, 2016). In brief, PGE$_2$ was extracted from homogenized tissue samples (*ca.* 25 mg) using liquid–liquid extraction. PGE$_2$ was analyzed using a Synergi Hydro column (150 × 2 mm, 4 μm, Phenomenex) coupled to a hybrid triple quadrupole-ion trap mass spectrometer QTRAP 5500 (Sciex, Darmstadt, Germany) equipped with a Turbo-V-source operating in negative ESI mode. The concentrations of the calibration standards, quality controls, and tissue samples were evaluated by Analyst software 1.6 and MultiQuant software 3.0 (Sciex) using the internal standard method (isotope dilution mass spectrometry). The calibration curve was calculated by quadratic regression with $1/x^2$ weighting (calibration range 0.002–25 ng/sample).

### RNA analysis

RNA was isolated from mouse ventricular tissue or from NRVMs using Trizol (Invitrogen). Total RNA was digested with DNase, and cDNA synthesis from 1,000 ng of RNA was carried out using a SuperScript first-strand synthesis system for RT–PCR (Invitrogen). Quantitative real-time PCR (qPCR) was performed with Universal ProbeLibrary (Roche) by using TaqMan Universal PCR Mastermix (Applied Biosystems) and detection on a 7500 Fast Cycler (Applied Biosystems). Primer sequences are shown in Appendix Table S2.

### *In vivo* assay for MEF2 activity

*In vivo* transcriptional activity of MEF2 was monitored using MEF2-LacZ reporter mice, as described previously (Naya *et al*, 1999; Passier *et al*, 2000). Briefly, the MEF2-driven β-galactosidase expression was determined by measuring the β-galactosidase activity.

### The paper explained

#### Problem

The underlying mechanisms responsible for the development of cardiac remodeling and myocardial dysfunction are still poorly understood. The aim of the study was to identify and describe new signaling pathways, leading to an activation of the transcription factor myocyte enhancer factor 2 (MEF2), a central regulator of adverse cardiac remodeling, and thereby potentially contributing to the pathogenesis of heart failure.

#### Results

We found that the inflammatory mediator prostaglandin E$_2$ (PGE$_2$) is a powerful activator of MEF2 in neonatal rat ventricular cardiomyocytes. In accordance with that, the levels of PGE$_2$ are increased and MEF2 is strongly activated in a murine model of sepsis *in vivo*. We provide evidence that the effects of PGE$_2$ are transmitted by the EP$_3$ receptor subtype and the βγ subunit of the heterotrimeric G$_{i/o}$ protein, which initiate at least two parallel signaling pathways. One consists of the guanine exchange factor Tiam1, the monomeric GTPase Rac1, and the p21-activated kinase 2. In addition, the other pathway results in the activation of protein kinase D, which relieves the histone deacetylase 5-mediated repression of MEF2 activity. Inhibition of either of the participating pathway members prevented the MEF2 activation by PGE$_2$.

#### Impact

Our findings reveal an unexpected new link between inflammation and adverse cardiac remodeling via MEF2 activation. The unmasked upstream signaling pathway consisting of PGE$_2$, EP$_3$, Rac1, and PKD may provide a number of new drug target candidates for the treatment of heart failure, in particular of inflammatory cardiomyopathies.

After LPS or saline treatment, β-galactosidase staining was performed on cardiac sections and whole organs, respectively, from MEF2-lacZ reporter mice. After fixation, β-galactosidase activity was detected by addition of the chromogenic substrate 5-bromo-4-chloro-3-indolyl-β-D-galactoside (X-gal), which results in creation of bright blue precipitates. Images were acquired with an Olympus SZH zoom stereo dissection scope with an Optronics DEI-750 CCD digital camera (using a 20× objective for histological slides); whole-heart stainings were quantified using ImageJ software by measuring the blue pixel intensity normalized to the total intensity.

### Statistics

Results are expressed as mean ± s.e.m. No statistical methods were used to predetermine the sample size. Sample size was chosen based on our own experience and other publications studying similar processes. Mice were randomly and blindly allocated to the sham-treated group and the LPS-treated group. Besides that, no randomization was used. Animals, which died in the first 24 h after treatment, were not further examined. Besides that, no animals or samples were excluded. To assess β-galactosidase expression and cell size of NRVMs, the investigator was blinded during both the image acquisition and the analysis. Analysis of HDAC5 cellular localization was blinded as well. For statistical analysis Student's or Welch's unpaired two-tailed tests, Pearson's correlation test, one-way or two-way ANOVA with Bonferroni *post hoc* tests were performed. D'Agostino-Pearson test was used to assess normal distribution. We considered *P*-values < 0.05 statistically significant.

Expanded View for this article is available online.

## Acknowledgements

We thank Claudia Heft, Michaela Oestringer, and Ulrike Oehl for technical assistance. GFP-HDAC5 adenovirus and MEF2-lacZ reporter mice were kindly provided by Eric N. Olson (University of Texas Southwestern Medical Center, Dallas, USA). An antibody against Ser259-phosphorylated HDAC5 was kindly provided by Timothy A. McKinsey (University of Colorado, Denver, USA). Furthermore, we thank Philip Eaton (University College London, London, UK) for providing advice concerning the LPS-induced endotoxemia. J.B. was supported by grants from the Deutsche Forschungsgemeinschaft (SFB 1118) and the Ministerium für Wissenschaft, Forschung, und Kunst Baden-Württemberg (Inflamyocard). J.B., T.W., M.L, and H.A.K were supported by the DZHK (Deutsches Zentrum für Herz-Kreislauf-Forschung—German Centre for Cardiovascular Research) and by the BMBF (German Ministry of Education and Research). A.D.T. was supported by the Jellinek Harry Scholarship (partnership between Heidelberg, Freiburg, and Semmelweis Universities). R.S. was supported by the DGK (Deutsche Gesellschaft für Kardiologie—German Cardiac Society) and the DZHK (Deutsches Zentrum für Herz-Kreislauf-Forschung— German Centre for Cardiovascular Research).

## Author contributions

ADT, RS, TW, and JB designed the study and analyzed the results; RS, CV, PT, CH, and FA performed animal experiments; ADT, RS, ML, CV, CH, SW, LF, and JK-H made cell culture experiments, DT performed mass spectrometric analysis; H-JG performed and analyzed histological stainings; MA provided important reagents; ADT, RS, ML, CV, PT, CH, FA, SW, LF, JK-H, DT, H-JG, MA, HAK, TW, and JB wrote the manuscript.

## Conflict of interest

The authors declare that they have no conflict of interest.

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
