## [Review Process File · EMBO Molecular Medicine]

Inflammation leads through PGE/EP3 signaling to HDAC5/MEF2-dependent transcription in cardiac myocytes

András D. Tóth, Richard Schell, Magdolna Lévy, Christiane Vettel, Philipp Theis, Clemens Haslinger, Felix Alban, Stefanie Werhahn, Lina Frischbier, Jutta Krebs-Haupenthal, Dominique Thomas, Hermann-Josef Gröne, Metin Avkiran, Hugo A. Katus, Thomas Wieland & Johannes Backs

Review timeline:

Submission date:	28 September 2017
Editorial Decision:	13 November 2017
Revision received:	31 March 2018
Editorial Decision:	7 May 2018
Revision received:	17 May 2018
Accepted:	18 May 2018

Editor: Céline Carret

Transaction Report:

1st Editorial Decision

13 November 2017

Thank you for the submission of your manuscript to EMBO Molecular Medicine. We have now heard back from the three referees whom we asked to evaluate your manuscript.

You will see from their comments pasted below, that the study is unanimously found important and solid. This said, all referees would like to see the *in vivo* data extended and more mechanism provided and good suggestions are offered to guide you along these lines should you decide to revise the paper.

As such, we would welcome the submission of a revised version within three months for further consideration and would like to encourage you to address all the criticisms raised as suggested to improve conclusiveness and clarity. Please note that EMBO Molecular Medicine strongly supports a single round of revision and that, as acceptance or rejection of the manuscript will depend on another round of review, your responses should be as complete as possible.

I look forward to receiving your revised manuscript.

***** Reviewer's comments *****

Referee #1 (Remarks for Author):

This is a very interesting and carefully designed study showing evidence that PGE2 activates MEF2 through a dual mechanism involving Gi/o beta-gamma-PKD and Rac1-Pak, using a combination of classical pharmacological approaches and adenovirus-mediated protein overexpression. Overall the data are solid and persuasive, as well as supporting earlier studies of the pro-hypertrophic effects of prostaglandins. The potential for PGE2 to contribute to a pro-inflammatory component of pathological hypertrophy is potentially of high importance.

Major comments:

1. Some limitations of the study include the lack of information about endogenous MEF2, specifically its activation state as determined by acetylation, its abundance, and any MEF2 isoform specificity of the effects. There is also no information about the impact of the various reagents on the myocytes themselves, including importantly their transcriptional responses. Did the authors note any instances of dissociation between MEF2 reporter activation and aspects of the myocyte hypertrophic response?
2. The authors use a number of pharmacological reagents at single concentrations- please provide support for the activity and specificity of the concentrations used for each, since this heavily impacts selectivity of action.
3. The authors state that adenylyl cyclase inhibition mimics the effect of *Gai/o*, and that it had no effect on MEF2 activation. They use this finding as well as transducin alpha overexpression to conclude that the $\beta\gamma$ subunit of *Gi/o* is responsible for MEF2 activation. While interesting, these results are not conclusive and should be supported by loss of function studies eg knockdown of the beta subunit.
4. It is not sufficient to phosphorylate HDAC5 to de-repress MEF2 as under some conditions pHDAC5 nuclear export can be blocked without derepression of MEF2 (Wei, Joshi et al JCI Insight 2017). Did the authors determine PKD and Pak1 effects on HDAC localization or MEF2 acetylation? It is intriguing to speculate that the Rac1-Pak pathway is acting directly on MEF2 rather than on its corepressor.
5. PKD has other targets in the cell, including p300- to what extent is p300 phosphorylation altered by PGE2 via PKD or via Rac1?
6. The authors may wish to address the seemingly conflicting finding that $G\beta\gamma$ hinders HDAC5 activity and how it affects interpretation of their results (their reference Spiegelberg & Hamm, 2005)

Referee #2 (Remarks for Author):

The authors addressed signaling mechanisms that control activity of the myocyte enhancer factor 2 (MEF2) transcription factor in cardiomyocytes. MEF2 has previously been shown to stimulate expression of genes that trigger pathological cardiac hypertrophy. Here, the authors show that prostaglandin E2 (PGE2) potently activates MEF2 in cardiomyocytes through two different mechanisms: (1) by triggering the $\beta\gamma$ subunit of *Gi/o*-proteins, which are activated downstream of the PGE2 receptor, EP3; (2) by protein kinase D (PKD)-mediated phosphorylation of class IIa HDACs such as HDAC5. These are convincing and important findings. Nonetheless, some additional experimentation would strengthen the manuscript, as detailed below.

Specific points

Major

1. The manuscript would be greatly enhanced by expansion of Figure 6. The stimulation of MEF2-driven beta-galactosidase expression by LPS is impressive but needs to be quantified (beta-galactosidase assay) from multiple Ns. Furthermore, the authors should perform immunoblotting to

determine the degree of PKD phosphorylation (Ser-916 and Ser-744/748) and HDAC5 phosphorylation in hearts of untreated vs. LPS treated mice. Finally, evaluation of the effects of inhibitors of the Tiam1/Rac1 pathway vs. PKD pathway on MEF2 activation in the heart in response to LPS would further strengthen the findings and provide mechanistic insights.

Minor

2. The first sentence of the Results section needs to be modified "To identify unknown GPCR-dependent signaling pathways, we conducted a screening experiment using neonatal rat ventricular cardiomyocytes (NRVMs)" should be changed to "...dependent signaling pathways that regulate MEF2 activity..."

3. In Fig. 1, the authors should assess other prostaglandins, such as PGF₂α, which has previously been shown to potently activate PKD in NRVMs.

4. In Fig. 3A, the authors should immunoblot for PKD (total, P-Ser-916, P-Ser-744/748).

5. The authors should reference the papers that originally described BPKDi so that readers can evaluate the compound's potency toward PKD isoforms and its selectivity for PKD over other kinases in the kinome.

Referee #3 (Comments on Novelty/Model System for Author):

As stated in the comments to the authors, it would be of great impact to the study if the authors would add some evidence that in an *in vivo* situation of endotoxemia, PKD and or PAK1 activity is enhanced as they show it in Figure 6 for MEF2. The study ends abruptly with showing expected increases in IL6 and TNF α mRNA expression, but it would substantially enhance the study if there would be data that show increased activity of the investigated pathways downstream of the EP3 receptor, in particular, as the authors seem to have this model up and running.

Referee #3 (Remarks for Author):

In their manuscript entitled: "Identification of a prostaglandin E-induced inflammatory pathway that activates the myocyte enhancer factor 2 in cardiac myocytes", Toth and colleagues investigate the molecular mechanisms that link elevation of pro-inflammatory mediators to MEF2-mediated cardiac remodeling. They describe the putative interplay of two different signalling axes in this scenario with Tiam1, Rac1, p21-activated kinase (PAK1) on one side and protein kinase D (PKD) leading to HDAC5 nucleo-cytoplasmic shuttling and finally MEF2-dependent gene transcription as the keyplayers and thus putative novel druggable targets.

The paper is nicely written and the experiments are carried out in a thoroughly controlled way. The content of the study is of translational relevance in diseases associated with chronic inflammation to avoid unwanted side effects associated with global inhibition of prostaglandin production.

Main concerns:

-In their study, the authors state that endothelin-1 leads to protein kinase C-independent activation of protein kinase D. This is in conflict with data published previously by the authors (Haworth et al. *JMCC* 2007;43:686-695). Can they please explain the apparent discrepancies to their previous findings (inhibition of specific PKC isoforms?) and provide a possible explanation? Why does abolished PKC-mediated Ser744/748 phosphorylation of PKD by exposure to pharmacological PKC inhibitors, does not affect the status of Ser916 phosphorylation?

-In Figure 5B it looks as if the authors detect two bands with the total PKD antibody, most likely reflecting PKD1 (upper band) and PKD2 (bottom band). Along those lines, it looks as if enhanced PGE₂-mediated Ser916 phosphorylation is mostly reflected by enhancement of the upper band - reflecting PKD1, whilst PKC-mediated Ser744/748 phosphorylation is mostly enhancing the bottom band. Can the authors please comment on this?

-PKD is very abundant in fibroblasts and the authors use neonatal rat ventricular myocytes, which contain also a substantial fibroblast content. Can the authors rule out whether PKD in fibroblasts

contributes (in part) to the observed responses to PGE₂? Is this pathway exclusively activated in cardiac myocytes or are there paracrine effects from factors released by fibroblasts acting on cardiac myocytes possible?

-PKD underlies a developmental decline in expression with low expression in the adult heart. Do the authors have evidence that the pathways they show to be important in neonatal cardiac myocytes also assume significance in the adult heart?

-The authors carefully index the activity status of PKD in response to PGE₂. Is there an equivalent activity readout for PAK1 activity available (phosphospecific antibody) that they can include?

- Figure 6 summarizes characteristic changes in proinflammatory cardiac gene transcription in response to LPS-induced endotoxemia. This is a somewhat an abrupt ending to this beautifully devised study. Can the authors include a read-out of PKD-and/or PAK1 activity as they have (at least in part) shown in cell culture before?

1st Revision - authors' response

31 March 2018

Referee #1 (Remarks for Author):

This is a very interesting and carefully designed study showing evidence that PGE₂ activates MEF2 through a dual mechanism involving Gi/o beta-gamma-PKD and Rac1-Pak, using a combination of classical pharmacological approaches and adenovirus-mediated protein overexpression. Overall the data are solid and persuasive, as well as supporting earlier studies of the pro-hypertrophic effects of prostaglandins. The potential for PGE₂ to contribute to a pro-inflammatory component of pathological hypertrophy is potentially of high importance.

Major comments:

1. Some limitations of the study include the lack of information about endogenous MEF2, specifically its activation state as determined by acetylation, its abundance, and any MEF2 isoform specificity of the effects. There is also no information about the impact of the various reagents on the myocytes themselves, including importantly their transcriptional responses. Did the authors note any instances of dissociation between MEF2 reporter activation and aspects of the myocyte hypertrophic response?

We thank the referee for these comments. In order to provide further evidence for the activation of endogenous MEF2, we investigated the impact of PGE₂ stimulation on the transcription of well-known MEF2 target genes. In good agreement with the results obtained with the 3xMEF2-Luc reporter, we found increased transcriptional response upon PGE₂ stimulation, reflected by the increased mRNA levels of *Nur77*, *Myomaxin* and *Adamts1* and the hypertrophy marker *BNP* (Fig. 1B of the revised manuscript).

As the referee pointed out, MEF2 activity could be enhanced independently from hypertrophic response of the heart. Thus, we performed new experiments to address this point: we found that PGE₂ stimulation led to MEF2 activation along with NRVM hypertrophy (Fig. 1C). The increase in cell size is congruent with the [³H]-leucine uptake results shown before (Fig. EV1B). Since MEF2 is not crucial for hypertrophy, we speculate that PGE₂ induces, in addition to MEF2, other pro-hypertrophic pathways as well.

We agree that it would also be interesting to identify the specific MEF2 isoforms involved in the PGE₂ effect. However, the experiments performed during the revision period point to a concept that both pathways, PKD and PAK2, converge on HDAC5, which in turn regulates all MEF2 isoforms. Moreover, we found during the revision that PGE₂ regulates MEF2 in adult cardiac myocytes as well, indicating that both MEF2A and MEF2D might be sensitive to PGE₂. Future studies using mice deficient for these two isoforms will unmask the relative contribution of the particular isoforms.

As suggested, we also started to examine the activation state of endogenous MEF2 upon PGE₂ treatment by exploring its post-translational modifications. Using the PhosTag system (see rebuttal Fig. 1), in which decreased electrophoretic mobility of the protein of interest indicates its increased

phosphorylation, we found no evidence of altered phosphorylation of MEF2 by using a MEF2D-specific antibody or an antibody recognizing all isoforms. As the referee also pointed out, MEF2 acetylation has a prominent role in the regulation of MEF2 activity. However, in our very preliminary experiments (co-immunoprecipitation of acetylated MEF2; see rebuttal Fig. 2) we could not find evidence for a change in its acetylation status but more experiments with more controls would be needed to come to a definite conclusion. However and importantly, the results of the experiments we performed to address the referee question 4 led us to the conclusion that PAK2 regulates nucleo-cytoplasmic shuttling of HDAC5. Thus, we modified the working model into the direction that HDAC5 serves as a point of convergence instead of MEF2. Therefore, we prefer not to include data into the manuscript that would imply or disprove (since the data presented here are still at a very preliminary stage) that PKD or PAK2 would result in post-translational modifications of MEF2.

Rebuttal Fig. 1:

Rebuttal Fig. 2:

2. The authors use a number of pharmacological reagents at single concentrations- please provide support for the activity and specificity of the concentrations used for each, since this heavily impacts selectivity of action.

The referee raised an important point. We show now a dose-response relationship of the PGE₂ effect on MEF2 activity (see Fig. EV1A). We also determined the half maximal inhibitory concentration of the EP₃ receptor antagonist L798106 (27 nM, K_i values are 0.3, 916, > 5000 and > 5000 nM at EP₃, EP₄, EP₁ and EP₂ receptors, respectively) (see Fig. EV1C). To show that L798106 inhibits indeed a cAMP-decreasing receptor of PGE₂, we assessed the phosphorylation of phospholamban at a PKA target site (Ser-16) (Cuello et al, 2007). In accordance, pretreatment with L798106 elevated the ISO-induced phosphorylation of phospholamban (see Fig. EV1D). These results support our conclusion that the PGE₂ effect is mediated by the EP₃ receptor.

The concentrations of all compounds, that were used in this study, were shown to act efficiently and with good specificity in previous studies and are widely-accepted for interrogation of signaling pathways. We also used these tools with great success in previous works. We added the supporting references to all compounds in Appendix Table 1.

3. The authors state that adenylyl cyclase inhibition mimics the effect of G_{i/o}, and that it had no effect on MEF2 activation. They use this finding as well as transducin alpha overexpression to conclude that the βγ subunit of G_{i/o} is responsible for MEF2 activation. While interesting, these results are not conclusive and should be supported by loss of function studies eg knockdown of the beta subunit.

We showed in this study that pertussis toxin, an extremely selective inhibitor of G_{i/o} proteins, inhibits the PGE₂-induced effect on MEF2 activity, which provides a proof for involvement of G_{i/o}. Although loss-of function studies can aid to underlie the effects of pharmacological reagents, we are afraid that this is not true in the current case. There are 4, 5, and 12 different genes coding α_{i/o}, β, and γ subunits, respectively. Furthermore, all β and γ subunits have to be knocked down combined to diminish the βγ-effect, which is not achievable technically. To dissect the involvement of G protein subunits in an interrogated pathway, overexpression of scavenger proteins is a widely-accepted way in pharmacological investigations, as we also did in this and previous studies. Overexpression of transducin α is a powerful tool to inhibit G-βγ, since the formed high-affinity complex of these proteins does not allow G-βγ to dissociate and transduce its signal downstream. We think that these findings underlie our claims convincingly. To further address the point of the referee, now we show that overexpression of RGS3L, another βγ-scavenger (Vogt et al, 2007), has the same effect as the overexpression transducin α (Fig. EV2).

4. It is not sufficient to phosphorylate HDAC5 to de-repress MEF2 as under some conditions pHDAC5 nuclear export can be blocked without derepression of MEF2 (Wei, Joshi et al JCI Insight 2017). Did the authors determine PKD and Pak1 effects on HDAC localization or MEF2 acetylation? It is intriguing to speculate that the Rac1-Pak pathway is acting directly on MEF2 rather than on its corepressor.

As the referee suggested, we examined the nuclear export of HDAC5 after PGE₂ treatment by overexpressing GFP-HDAC5 (Fig. 6). Similar to ET1, PGE₂ induced remarkable translocation of HDAC5 from the nucleus to the cytosol. The PGE₂-induced translocation of HDAC5 was prevented both by inhibitors of PKD and PAK. These results suggest that both PKD and the Tiam1-Rac1-PAK2 pathways converge on HDAC5 and participate in the regulation of nucleo-cytoplasmic shuttling of HDAC5. Therefore, we conclude that nuclear export of HDAC5 is a major mechanism of PGE₂-induced MEF2-activation, although we cannot completely rule out that other additive mechanisms of MEF2 activation, such as post-translational modifications of MEF2, contribute also to MEF2 activation. This is now discussed in the revised manuscript in more detail: “*The necessity of multiple signaling pathways for the complete activation of MEF2 is in good agreement with the results of previous studies. For instance, nuclear export of HDACs can be blocked without altering their phosphorylation, and acetylation of MEF2 was shown to be essential for its complete activation (Wei et al, 2017).*”

5. PKD has other targets in the cell, including p300- to what extent is p300 phosphorylation altered by PGE2 via PKD or via Rac1?

According to the referee’s suggestions, we examined the phosphorylation state of p300 in NRVMs. At the examined phosphorylation site (Ser-1834), we found no difference of p300 phosphorylation after 1-hour treatment with PGE₂ (see rebuttal Fig. 3). However, this pathway may be also activated,

because we found – as expected - increased phosphorylation of CREB at a PKD target site (Ser-133) (Ozgen et al, 2008), which protein is an important co-activator of p300 and CBP (see rebuttal Fig. 4). Nonetheless, our preliminary results did not point to obvious changes in acetylation of endogenous MEF2 (see rebuttal Fig. 2).

Rebuttal Fig. 3:

Rebuttal Fig. 4:

6. The authors may wish to address the seemingly conflicting finding that G β γ hinders HDAC5 activity and how it affects interpretation of their results (their reference Spiegelberg & Hamm, 2005)

Our results show that PKD-mediated HDAC5 inhibition is essential in the PGE₂-induced MEF2 activation. Interestingly, in the study of Spiegelberg & Hamm it was shown that G-βγ can also hinder HDAC5 function and derepress MEF2. Thus, the G-βγ-mediated inhibition of HDAC5 could be an interesting additive mechanism of depressing HDAC5 function. This is now discussed in more detail in the manuscript: “In addition, Gβγ was also shown to hinder HDAC5 activity through direct binding, which might add to the effects described in this study (Spiegelberg & Hamm, 2005)”

Referee #2 (Remarks for Author):

The authors addressed signaling mechanisms that control activity of the myocyte enhancer factor 2 (MEF2) transcription factor in cardiomyocytes. MEF2 has previously been shown to stimulate expression of genes that trigger pathological cardiac hypertrophy. Here, the authors show that prostaglandin E2 (PGE2) potently activates MEF2 in cardiomyocytes through two different mechanisms: (1) by triggering the βγ subunit of Gi/o-proteins, which are activated downstream of the PGE2 receptor, EP3; (2) by protein kinase D (PKD)-mediated phosphorylation of class IIa HDACs such as HDAC5. These are convincing and important findings. Nonetheless, some additional experimentation would strengthen the manuscript, as detailed below.

Major

The manuscript would be greatly enhanced by expansion of Figure 6. The stimulation of MEF2-driven beta-galactosidase expression by LPS is impressive but needs to be quantified (beta-galactosidase assay) from multiple Ns. Furthermore, the authors should perform immunoblotting to determine the degree of PKD phosphorylation (Ser-916 and Ser-744/748) and HDAC5 phosphorylation in hearts of untreated vs. LPS treated mice. Finally, evaluation of the effects of inhibitors of the Tiam1/Rac1 pathway vs. PKD pathway on MEF2 activation in the heart in response to LPS would further strengthen the findings and provide mechanistic insights.

We appreciate the referee's comment. We added quantification of the MEF2-driven β -galactosidase expression by LPS (Fig. 7C). Furthermore, we performed immunoblotting from heart samples of saline and LPS treated mice, and found marked increase in PKD, HDAC5 and PAK phosphorylation after LPS treatment (Fig. 7D in the revised manuscript). These results underlie the activation of these pathways in myocardial inflammation *in vivo*.

We absolutely agree with the referee that the treatment of LPS mice with the mentioned inhibitors would provide more *in vivo* relevance and further mechanistic insights. However, these experiments were not possible in the given time for the revision and are, therefore, beyond the scope of the current study.

Minor

2. The first sentence of the Results section needs to be modified "To identify unknown GPCR-dependent signaling pathways, we conducted a screening experiment using neonatal rat ventricular cardiomyocytes (NRVMs)" should be changed to "...dependent signaling pathways that regulate MEF2 activity ..."

We modified the sentence according to the referee's suggestions.

3. In Fig. 1, the authors should assess other prostaglandins, such as PGF₂ α , which has previously been shown to potently activate PKD in NRVMs.

In our screening we also assessed other prostaglandins using metabolically stable analogs (Fig. 1A). Neither fluprostenol (PGF_{2 α} analog), nor trepostinil (prostacyclin receptor agonist) were able to induce MEF2 activity. The lack of effect of fluprostenol was surprising, as we and others have shown pro-hypertrophic and PKD-activating effects of FP receptor stimulation. Possible explanations could be that PGF_{2 α} may induce a repressing pathway or activate not all the crucial pathways for MEF2 derepression.

4. In Fig. 3A, the authors should immunoblot for PKD (total, P-Ser-916, P-Ser-744/748).

According to the referee's suggestions, we assessed PKD phosphorylation upon pretreatment with kinase inhibitors. Please find our comments on the results in the answer to the first and second questions of referee #3.

5. The authors should reference the papers that originally described BPKDi so that readers can evaluate the compound's potency toward PKD isoforms and its selectivity for PKD over other kinases in the kinome.

We thank the referee for this advice, and we added these references.

Referee #3

As stated in the comments to the authors, it would be of great impact to the study if the authors would add some evidence that in an *in vivo* situation of endotoxemia, PKD and or PAK1 activity is enhanced as they show it in Figure 6 for MEF2. The study ends abruptly with showing expected increases in IL6 and TNF alpha mRNA expression, but it would substantially enhance the study if there would be data that show increased activity of the investigated pathways downstream of the EP3 receptor, in particular, as the authors seem to have this model up and running.

We thank the referee for this comment. According to this suggestion (which is similar to point 1 of referee 2), we have expanded our *in vivo* data with the investigation of the described pathways. We added quantification of the MEF2-driven β -galactosidase expression in MEF2 reporter mice (Fig. 7C). Moreover, we performed immunoblotting to assess the activity of the pathways downstream to EP₃ receptor. We found pronounced increase in PKD, HDAC5 and PAK phosphorylation in LPS-treated mice (Fig. 7D). These results underlie the activation of these pathways in myocardial inflammation *in vivo*.

In their manuscript entitled: "Identification of a prostaglandin E-induced inflammatory pathway that activates the myocyte enhancer factor 2 in cardiac myocytes", Toth and colleagues investigate the molecular mechanisms that link elevation of pro-inflammatory mediators to MEF2-mediated cardiac remodeling. They describe the putative interplay of two different signalling axes in this scenario with Tiam1, Rac1, p21-activated kinase (PAK1) on one side and protein kinase D (PKD) leading to HDAC5 nucleo-cytoplasmic shuttling and finally MEF2-dependent gene transcription as the keyplayers and thus putative novel druggable targets. The paper is nicely written and the experiments are carried out in a thoroughly controlled way. The content of the study is of translational relevance in diseases associated with chronic inflammation to avoid unwanted side effects associated with global inhibition of prostaglandin production.

Main concerns:

-In their study, the authors state that endothelin-1 leads to protein kinase C-independent activation of protein kinase D. This is in conflict with data published previously by the authors (Haworth et al. JMCC 2007;43:686-695). Can they please explain the apparent discrepancies to their previous findings (inhibition of specific PKC isoforms?) and provide a possible explanation? Why does abolished PKC-mediated Ser744/748 phosphorylation of PKD by exposure to pharmacological PKC inhibitors, does not affect the status of Ser916 phosphorylation?

As Referee #2 suggested, we assessed PKD phosphorylation upon pretreatment with the PKC-inhibitor BIM (Fig. 3B). Interestingly, the ET1-induced phosphorylation of PKD was hindered by BIM at Ser-744/Ser-748 but not altered at Ser-916, showing that ET1 stimulation leads both to PKC-dependent and -independent activation of PKD. In the work of (Haworth et al, 2007), the role of PKC in activation of PKD was investigated by examining the phosphorylation at Ser-744/Ser-748, a known PKC target site. BIM (or also known as GF109203X), similar to other PKC inhibitors, markedly decreased the phosphorylation at Ser-744/Ser-748, similar to the results of this study. However, in the work of Haworth et al, Ser-916 phosphorylation was not investigated, which seems to be regulated by PKC-independent mechanisms (see also the answer to the next question). ET-1 has been shown to lead to PKC-independent activation of PKD in previous studies by other groups (Guo et al, 2011; Vega et al, 2004). Notably, those studies and the present work were performed in neonatal rat cardiac myocytes, whereas the study of Haworth et al. was performed in adult rat cardiac myocytes. It is possible that the relative contribution of PKC to ET-1-induced PKD activation, as reflected by autophosphorylation at Ser-916, may vary depending on the differentiation stage of the cardiac myocytes used.

-In Figure 5B it looks as if the authors detect two bands with the total PKD antibody, most likely reflecting PKD1 (upper band) and PKD2 (bottom band). Along those lines, it looks as if enhanced PGE2-mediated Ser916 phosphorylation is mostly reflected by enhancement of the upper band - reflecting PKD1, whilst PKC-mediated Ser744/748 phosphorylation is mostly enhancing the bottom band. Can the authors please comment on this?

As the referee pointed out, the Ser-744/Ser-748 residues of PKD are generally considered as PKC target sites. In accordance, pretreatment with BIM abolished the phosphorylation of these sites upon PGE₂ (Fig. 3B in the revised manuscript), but the autophosphorylation (at Ser-916) was only slightly altered. In contrast to BIM, the PKD inhibitor BPKDi prevented only the autophosphorylation, but had no effect on Ser-744/Ser-748 phosphorylation. These results show that PGE₂ activates PKD both via PKC-dependent and -independent mechanisms. It would be intriguing to speculate that the PKD1 isoform can be activated PKC-independently, whereas PKD2 only through PKC. However, since phosphorylation can alter the electrophoretic mobility of the protein of interest, we think that this conclusion cannot be made based only on these experiments. We plan to conduct future in vivo studies using mice deficient for different PKD isoforms. These experiments may answer the very interesting point raised by the referee.

-PKD is very abundant in fibroblasts and the authors use neonatal rat ventricular myocytes, which contain also a substantial fibroblast content. Can the authors rule out whether PKD in fibroblasts contributes (in part) to the observed responses to PGE2? Is this pathway exclusively activated in cardiac myocytes or are there paracrine effects from factors released by fibroblasts acting on cardiac myocytes possible?

The referee raised an important point. To answer this, we investigated whether PKD or PAK are activated in primary cardiac fibroblasts (Fig. 5C in the revised manuscript) after PGE₂ stimulation. We found no signal of PKD or PAK2 activation in fibroblasts after PGE₂ treatment. These results suggest that the observed responses rather originate from cardiac myocytes.

-PKD underlies a developmental decline in expression with low expression in the adult heart. Do the authors have evidence that the pathways they show to be important in neonatal cardiac myocytes also assume significance in the adult heart?

As the referee pointed out, it has been demonstrated that PKD expression decreases with age. However, significant levels of PKD can still be detected in hearts of adult mice (Fig. 7C). Moreover, we found increased phosphorylation of PKD and PAK2 in LPS-treated adult mice, showing that the PGE₂-induced pathways are also activated in cardiac inflammation in adults. Furthermore, PKD phosphorylation is also promoted by PGE₂ in adult murine cardiac myocytes (Fig. EV3), supporting again that the suggested pathway is relevant in adults. However, the adult data suggest that the PKC-dependent mode of PKD activation may be more relevant as compared to the neonatal situation.

-The authors carefully index the activity status of PKD in response to PGE2. Is there an equivalent activity readout for PAK1 activity available (phosphospecific antibody) that they can include?

According to the referee's suggestions, we assessed PAK phosphorylation in NRVMs (Fig. 5C in the revised manuscript). From the three existing PAK isoforms, PAK2 was found to be activated after PGE₂ stimulation of NRVMs, but the activation is absent in cardiac fibroblasts (see the question regarding to fibroblasts). In addition, some mice responded with increased PAK2 phosphorylation upon LPS-treatment, which was observed in none of the sham-treated mice. The increase was even statistically significant in the case of Thr-402 phosphorylation of PAK2.

- Figure 6 summarises characteristic changes in proinflammatory cardiac gene transcription in response to LPS-induced endotoxemia. This is a somewhat an abrupt ending to this beautifully devised study. Can the authors include a read-out of PKD-and/or PAK1 activity as they have (at least in part) shown in cell culture before?

According to all referees' suggestions, we extended the *in vivo* results with PKD, PAK and HDAC5 phosphorylation data for the LPS model (Fig. 7D).

References

- Cuello F, Bardswell SC, Haworth RS, Yin X, Lutz S, Wieland T, Mayr M, Kentish JC, Avkiran M (2007) Protein kinase D selectively targets cardiac troponin I and regulates myofilament Ca²⁺ sensitivity in ventricular myocytes. *Circulation research* 100: 864-873
- Guo J, Gertsberg Z, Ozgen N, Sabri A, Steinberg SF (2011) Protein kinase D isoforms are activated in an agonist-specific manner in cardiomyocytes. *The Journal of biological chemistry* 286: 6500-6509
- Haworth RS, Roberts NA, Cuello F, Avkiran M (2007) Regulation of protein kinase D activity in adult myocardium: novel counter-regulatory roles for protein kinase Cepsilon and protein kinase A. *Journal of molecular and cellular cardiology* 43: 686-695
- Ozgen N, Obrezhtchikova M, Guo J, Elouardighi H, Dorn GW, 2nd, Wilson BA, Steinberg SF (2008) Protein kinase D links Gq-coupled receptors to cAMP response element-binding protein (CREB)-Ser133 phosphorylation in the heart. *The Journal of biological chemistry* 283: 17009-17019
- Vega RB, Harrison BC, Meadows E, Roberts CR, Papst PJ, Olson EN, McKinsey TA (2004) Protein kinases C and D mediate agonist-dependent cardiac hypertrophy through nuclear export of histone deacetylase 5. *Molecular and cellular biology* 24: 8374-8385

Vogt A, Lutz S, Rumenapp U, Han L, Jakobs KH, Schmidt M, Wieland T (2007) Regulator of G-protein signalling 3 redirects prototypical Gi-coupled receptors from Rac1 to RhoA activation. *Cellular signalling* 19: 1229-1237

Wei J, Joshi S, Speransky S, Crowley C, Jayathilaka N, Lei X, Wu Y, Gai D, Jain S, Hoosien M et al (2017) Reversal of pathological cardiac hypertrophy via the MEF2-coregulator interface. *JCI Insight* 2

2nd Editorial Decision

7 May 2018

Thank you for the submission of your revised manuscript to EMBO Molecular Medicine. We have now received the enclosed reports from the referees that were asked to re-assess it. As you will see the reviewers are now globally supportive and I am pleased to inform you that we will be able to accept your manuscript pending the following final amendments:

1) Please address referee 1's comments.

At this stage, we'd like you to discuss referee's 1 points and if you do have data at hand, we'd be happy for you to include it, however we will not ask you to provide any additional experiments at this stage. The point raised that HDAC5 is more the focus of the paper rather than MEF2 is an important point and we'd like to see your view about it, including a potential title & abstract change to reflect this emphasis.

Please provide a letter INCLUDING my comments and the reviewer's reports and your detailed responses to their comments (as Word file).

Please submit your revised manuscript within two weeks. I look forward to seeing a revised form of your manuscript as soon as possible.

***** Reviewer's comments *****

Referee #1 (Remarks for Author):

The authors have replied with new experiments and a thoughtful reinterpretation of their data, indicating that the target of PGE2 signalling to MEF2 is HDAC5 nuclear localization. I remain very enthusiastic about this careful study. However I feel that the new data take MEF2 somewhat out of the forefront. Specifically, there isn't any evidence that MEF2 itself is the target.

1. The fact that a MEF2 reporter gene is expressed at higher levels in the presence of PGE2 is not sufficient evidence for a specific effect on MEF2 as opposed to an effect on one or more cofactors or co-activators.

2. In agreement with the latter, their new data show that HDAC5 is phosphorylated in LPS-treated mice and that its nuclear export is regulated by PGE2 downstream of two distinct effectors, PAK and PKD. This important observation places HDAC5 at the common terminus of both pathways.

2. Other new data suggest that there is no change in bulk MEF2 modification by phosphorylation or acetylation by PGE2. This is surprising since the impact of HDAC export is to permit acetylation and activation of TFs such as MEF2 that are otherwise silenced. It appears that the change is either too small to detect or non-existent. A case could be made that MEF2 is not the most important target of the PGE2 signal.

3. The authors provide evidence that other factors could be targeted by PGE2 to drive hypertrophy, specifically CREB. One could argue that a small or promoter-specific change in active MEF2 is sufficient to explain the increase in MEF2 reporter gene expression. But this data is also consistent with the possibility that reporter gene activation involves other, unidentified factors that are more directly regulated by HDAC5 and/or PKD and PAK. The authors' comment that "MEF2 is not crucial for hypertrophy" indicates that they have considered this possibility.

To summarize, I agree with the authors that the role of MEF2 activation through covalent modification (as it is generally understood to occur) in this model is unclear. Accordingly I think the HDAC5 focus is appropriate, stands by itself, and should be the primary focus of the title and the discussion rather than MEF2.

Referee #2 (Remarks for Author):

The authors have done an excellent job of addressing my comments.

2nd Revision - authors' response

17 May 2018

Referee 1:

“The authors have replied with new experiments and a thoughtful reinterpretation of their data, indicating that the target of PGE2 signalling to MEF2 is HDAC5 nuclear localization. I remain very enthusiastic about this careful study. However I feel that the new data take MEF2 somewhat out of the forefront. Specifically, there isn't any evidence that MEF2 itself is the target.

1. The fact that a MEF2 reporter gene is expressed at higher levels in the presence of PGE2 is not sufficient evidence for a specific effect on MEF2 as opposed to an effect on one or more cofactors or co-activators.”

We thank the referee for the enthusiastic feedback. We agree that the revision shed more light on HDAC5 and not on direct posttranslational modifications of MEF2. However, we used a MEF2 reporter for the initial screen, MEF2 activation was the readout throughout the manuscript and we also included MEF2 reporter mice to demonstrate *in vivo* activation of MEF2 upon inflammation. Therefore, we would still like to mention MEF2 in the title. We propose to add HDAC5 to the title to indicate that MEF2 regulation is due to HDAC5 regulation. Therefore, we changed the title to: “Inflammation leads through PGE/EP₃ signaling to HDAC5/MEF2-dependent transcription in cardiac myocytes”

Moreover, in the abstract we conclude that “our findings provide an unexpected new link between inflammation and cardiac remodeling by derepression of MEF2 through HDAC5 inactivation, ...” to emphasize the part of HDAC5 more clearly.

2. Other new data suggest that there is no change in bulk MEF2 modification by phosphorylation or acetylation by PGE2. This is surprising since the impact of HDAC export is to permit acetylation and activation of TFs such as MEF2 that are otherwise silenced. It appears that the change is either too small to detect or non-existent. A case could be made that MEF2 is not the most important target of the PGE2 signal.

We agree with the referee that HDAC5 export permits MEF2 acetylation. We cannot finally be sure whether we were unable to detect MEF2 acetylation because of technical limitations or for other reasons. However, we are sure that PGE signaling has a profound effect on MEF2 activation, in particular when compared to other GPCR agonists. We agree that PGE2 has other targets, but the focus of this study was MEF2 activation.

3. The authors provide evidence that other factors could be targeted by PGE2 to drive hypertrophy, specifically CREB. One could argue that a small or promoter-specific change in active MEF2 is sufficient to explain the increase in MEF2 reporter gene expression. But this data is also consistent with the possibility that reporter gene activation involves other, unidentified factors that are more directly regulated by HDAC5 and/or PKD and PAK. The authors' comment that "MEF2 is not crucial for hypertrophy" indicates that they have considered this possibility.

To summarize, I agree with the authors that the role of MEF2 activation through covalent modification (as it is generally understood to occur) in this model is unclear. Accordingly I

think the HDAC5 focus is appropriate, stands by itself, and should be the primary focus of the title and the discussion rather than MEF2.

As discussed above, we changed the title and abstract accordingly.

Corresponding Author Name: Prof. Dr. med. Johannes Backs

Manuscript Number: EMM-2017-08536